# Problems with using comparative analyses of avian brain size to test hypotheses of cognitive evolution

Rebecca Hooper[1,2]*, Becky Brett[1], Alex Thornton[1]*

**1** University of Exeter, Centre for Ecology and Conservation, College of Life and Environmental Sciences, Penryn Campus, Cornwall, United Kingdom, **2** University of Exeter, Centre for Research in Animal Behaviour, College of Life and Environmental Sciences, Streatham Campus, Exeter, United Kingdom

* r.c.hooper2@exeter.ac.uk (RH); alex.thornton@exeter.ac.uk (AT)

## Abstract

There are multiple hypotheses for the evolution of cognition. The most prominent hypotheses are the Social Intelligence Hypothesis (SIH) and the Ecological Intelligence Hypothesis (EIH), which are often pitted against one another. These hypotheses tend to be tested using broad-scale comparative studies of brain size, where brain size is used as a proxy of cognitive ability, and various social and/or ecological variables are included as predictors. Here, we test how robust conclusions drawn from such analyses may be. First, we investigate variation in brain and body size measurements across >1000 bird species. We demonstrate that there is substantial variation in brain and body size estimates across datasets, indicating that conclusions drawn from comparative brain size models are likely to differ depending on the source of the data. Following this, we subset our data to the Corvides infraorder and interrogate how modelling decisions impact results. We show that model results change substantially depending on variable inclusion, source and classification. Indeed, we could have drawn multiple contradictory conclusions about the principal drivers of brain size evolution. These results reflect concerns from a growing number of researchers that conclusions drawn from comparative brain size studies may not be robust. We suggest that to interrogate hypotheses of cognitive evolution, a fruitful way forward is to focus on testing cognitive performance within and between closely related taxa, with an emphasis on understanding the relationship between informational uncertainty and cognitive evolution.

## Introduction

The principal drivers of cognitive evolution have been debated for decades [1–8]. Researchers often fall into two broad camps, focusing primarily on either social or ecological factors. Briefly, the Social Intelligence Hypothesis (SIH) posits that cognitive evolution is principally driven by the informational challenges of navigating a dynamic social environment, such as the need to track, anticipate and respond to the behaviour of social partners, and monitor the relationships of others [6, 7, 9–11]. In contrast, the Ecological Intelligence Hypothesis (EIH) and its variants emphasise informational challenges posed by ecological variables, such as

**Data Availability Statement:** Data and R script can be accessed at this link: 10.6084/m9.figshare.17081297.

**Funding:** This work was funded by a Natural Environment Research Council GW4 studentship (grant no. NERC 107672G) (https://nerc.ukri.org/) to RH and a Leverhulme grant (grant no. RGP-2020-170) (https://www.leverhulme.ac.uk/) to AT. The funders had no role in study design, data collection and analysis, decision to publish, or preparation of the manuscript.

**Competing interests:** The authors have declared that no competing interests exist.

changeable food sources and climatic conditions [2, 8, 12–15]. A large body of research has investigated social and ecological correlates of brain size across a range of taxa, including primates, ungulates, carnivores and birds [3, 16–28]. However, results are often inconsistent and contradictory [29–32]. For instance, Dunbar (1992) [3], found that primate social group size positively correlated with a measure of brain size, which is commonly used as a proxy for cognitive ability. In contrast, DeCasien et al. (2017) [17] found that diet is an important driver of primate brain size but social group size is not. Wartel et al. (2019) [32], on the other hand, found that either diet or social group size could predict primate brain size, depending on where brain size data and predictor variables were sourced from. Although the majority of research interrogating the SIH and EIH has focused on primates, birds have emerged as a major model system in cognitive evolution over the last 20 years (eg. [21, 33–35]). Some species of bird show convergent cognitive performance to primates [33, 35], yet birds have divergent neuroanatomy [33] and differing constraints on brain size, such as those imposed by long-range migration [36]. Here we interrogate the potential pitfalls that arise in the comparative study of cognitive evolution in birds. Moreover, we highlight potential pitfalls of current methodologies that have not yet been investigated in any taxa.

The relationship between brain size and cognitive ability is largely unknown and highly contentious [29, 30, 37, 38]. Nevertheless, studies investigating comparative cognitive evolution very often use some measure of brain size as a proxy of cognitive ability [32]. Most comparative studies of brain size use one measurement of brain size per species, taken either from a single individual or averaged across multiple individuals (e.g. [16, 18, 25, 39–43]). How much estimates differ between datasets, and whether this may influence the conclusions drawn from comparative analyses of brain size, are, however, poorly understood [31]. Moreover, to control for the relationship between brain and body size, most studies of brain size control for body size ('relative brain size') [30]. However, although body size measures may have important implications for the conclusions of comparative analyses brain size [44], the degree to which body size estimates varies between published datasets, and how much this may influence results, has yet to be considered beyond primates [44].

The approach that researchers take toward model specification also has the potential to drastically influence results. While most studies utilise similar statistical techniques to test hypotheses of brain size evolution, approaches toward model specification differ. Indeed, some researchers opt to include covariates associated only with the hypothesis of interest (broadly, the SIH or EIH), and either omit (e.g. [43]) or include less detailed (e.g. [16, 21]) variables associated with the competing hypothesis. However, the combination of variables is known to have a substantial influence on the results of primate and carnivora brain size models [27, 32]. In addition, where covariates are sourced from can have a substantial impact on results. For instance, Wartel et al. (2019) [32] showed that changing the source from which covariates (e.g. diet) were collected substantially changed the results of a previous study [17]. Moreover, decisions on how to define variables can be a somewhat subjective decision made by authors, and this may also have a significant influence on results. For example, if some populations of a species are migratory, but most are resident (e.g. as in the jackdaw, *Corvus monedula*, where only northern and eastern European populations migrate [45]), should the species be classified as migrant, resident, or a different category altogether? How such classification decisions influence model results is, as yet, unquantified.

In this study, we collate data from multiple datasets of brain and body size, which span more than 1000 bird species, to quantify the degree to which estimates vary across datasets (Aim 1). Given that most comparative studies of brain size use only one estimate of brain and body size, variation in estimates has the potential to substantially change results. Second, we use detailed data from the well-studied Corvides infra-order to interrogate whether

conclusions drawn from models testing alternative hypotheses for cognitive evolution differ depending on the combination, source and classification of variables included (Aim 2). Together, these investigations allow us to both identify novel pitfalls in the study of comparative cognition, and highlight parallel pitfalls to those previously identified in the field of primate comparative cognition [31, 32, 44].

## Methods

### Aim 1—Quantifying variation in brain and body size between datasets

Here, we use multiple datasets of brain and body size to investigate variation in estimates for more than 1000 bird species. Whole brain volumes across bird species were collated from six published datasets [18, 21, 46–49], all of which recorded brain volume using either the endocranial volume technique (see [50] for details), brain mass converted to volume [50], or both. These datasets are non-independent, with some measurements shared between them. Each dataset reported one datapoint per species, with the exception of [47] which reported two datapoints per species (one female, one male) and their associated standard errors (where more than one specimen per sex was used).

　Body mass was collated from eleven published datasets from ten studies [18, 21, 39, 46, 48, 51–55], all of which investigated brain size evolution. All body masses were measured in grams. One study collated two independently collected sources of body mass data [54] and these were thus treated as two different datasets. Again, datasets are not independent, with some using overlapping sources. Each dataset (aside from [54]) had one datapoint per species; one dataset contained standard errors associated with the estimate [51].

### Aim 2—Influence of variable inclusion, classification and source on model results

For this analysis, we collated detailed social and ecological variables for species in the Corvides infraorder. The Corvides, a diverse clade of passerines, originated ~30 million years ago in the proto-Papuan archipelago, and have since spread to all continents except Antarctica [56]. The Corvides are well-suited to this investigation because they have a well-resolved phylogeny [56], large variation in brain size [18], and many species have known social and ecological variables (see S4 Dataset). Brain and body size estimates were often identical between datasets, indicating measurements are shared between datasets or may come from the same individual. Given the non-independence of datasets, we were unable to analyse the impact that using independent datasets would have on study results.

　We tested whether including detailed ecological *and* social covariates, relative to including ecological *or* social covariates, qualitatively changed conclusions of models. To do this, we extracted/collated detailed ecological and social variables that have previously been shown to have a significant relationship with brain size (see *Methods*: *Variables*). In addition to constructing models with differing sets of predictor variables, we examined how sensitive model results were to choices regarding the classification of variables (see *Methods*: *Variables*: *Re-classification*). We also tested whether collecting variables from differing sources changed model results (see *Methods*: *Variables*: *Environmental variation*).

### Variables

We extracted/collated the following detailed ecological and social variables that have previously been shown to have a significant relationship with brain size.

**1. Ecological variables.** We included species movement, environmental variability and diet. Species that migrate are thought to have smaller brains than resident species [36, 42, 55, 57]. This is hypothesised to be because the energetic cost of the brain constrains selection on increased brain size in migrating species, who have large energetic demands during migration [36, 55, 57]. Meanwhile, species that live in fluctuating environments [21, 39, 58] and species with broader diets [21] are thought to have bigger brains than those in more stable environments or with specialist diets.

*1.1 Movement*. We coded species movement using four categories: resident, partial migrant, migrant or nomadic. Previous studies including migration as a covariate tend to include migration as a binary variable (resident or migratory: [39, 42]). However, some species are only migratory in certain regions (e.g. jackdaws). Such species were therefore coded as partial migrants (but see *Methods*: *Variables*: *Re-classification)*. Meanwhile, other species are neither migrants (all populations fly long-distance to a new home range seasonally), nor partial migrants (some populations fly long-distance to a new home seasonally) nor residents (remain at the home range year-round) and can instead be considered nomadic (no clear home range).

*1.2 Environmental variability*. We collected environmental variability from two sources. The first measure of environmental variability was 'temperature variation', as reported in Fristoe et al. (2017) [39], where higher values indicate more variability. The second was a measure of environmental variability calculated by Sayol et al. (2016) [21]. Briefly, Sayol et al. included multiple environmental variables in a phylogenetic principal component analysis. The resultant phylogenetic principal component 1 (PPC1) captured seasonal variation, duration of snow cover and among-year variation, with higher values indicating higher variation. PPC1 can therefore be interpreted as an axis describing general environmental variation. Meanwhile, phylogenetic principal component 2 (PPC2) captured environmental variation at lower latitudes (e.g., drought events). Temperature variation and PPCs were never used in the same models; instead, they were interpreted as two independent sources of 'environmental variation', which we used to quantify whether differing variable source may influence results.

*1.3 Diet breadth*. We used diet breadth as reported in Sayol et al. (2016) [21] who used Rao's quadratic entropy [59] with diet frequency for seven diet types (carrion, fruit, invertebrates, nectar, seed, vertebrates and plants). The range of values in our sample was 0–0.23, with higher values indicating a broader diet [21].

**2. Social variables.** We used two social variables in our models, both of which have been suggested to be involved in brain size evolution: social foraging [43, 60], where individuals forage in groups rather than solitarily, and cooperative breeding [18, 61], where group members help to care for offspring that are not their own [62]. While long-term monogamy has been shown to positively correlate with brain size in some studies [42, 43], almost all species in our sample form long-term monogamous pair bonds (see S4 Dataset) so there was not enough variation for this variable to be included.

*2.1 Social foraging*. Foraging group structure has previously been shown to correlate with relative brain size [42]. Specifically, species that forage in pairs or bonded groups have been shown to have larger brains than those that forage in large aggregations [42]. Similarly, species that live in small groups have been shown to have bigger brains than those that live in large aggregations [43]. This is argued to be because the *quality* rather than *quantity* of social bonds is a key driver of cognitive evolution in birds [42, 43]. However, foraging group structure appears to be unimportant in other studies [21]. A common problem with the inclusion of social variables in comparative studies is that they may not capture the underlying informational demands which, according to the SIH, drive cognitive evolution [10, 63, 64]. We therefore expanded on previous categorisations of foraging group structure by trying to capture variables thought to be associated with information-processing. Specifically, species were

coded as foraging solitarily, in pairs, in small groups (<30 individuals), in aggregations (>30 individuals), or as nested versions of these variables (e.g., forages in pairs nested within larger groups). The threshold of 30 individuals between small groups and aggregations was used following Sayol et al. (2016) [21]. If a species is known to forage in different social contexts but not necessarily in a nested fashion, we categorised these species using the largest group size commonly recorded (e.g., if the species forages in pairs and in small groups, but not necessarily in a nested manner, we recorded this as small group foraging).

*2.2 Cooperative breeding*. The role of cooperative breeding in cognitive evolution is contentious. Some authors argue that cooperative breeding entails substantial cognitive demands because individuals need to cooperate and coordinate with multiple others to raise offspring [61, 65, 66]. Conversely, others suggest that the typically high levels of relatedness and shared interests within cooperatively breeding groups may in fact reduce cognitive demands relative to independent breeding [64, 67, 68]. Relevant empirical evidence remains limited and controversial. For instance, Burkart & van Schaik (2009) [61] suggest that cooperatively breeding monkeys show elevated socio-cognitive performance, but these species also have particularly small brains [62], and rank poorly in meta-analyses of cognitive performance across primates [69]. Among birds, the only comparative study to date found no relationship between cooperative breeding and brain size [18], but this study did not include variables since shown to be significantly related to brain size, such as diet and environmental variation [21]. We therefore included cooperative breeding as a binary variable in our analyses. We note that species such as American crows (*Corvus brachyrhynchos*) and carrion crows (*Corvus corone*) are facultative cooperative breeders, but as there were few species in our sample that could be defined as such, we classified all facultative cooperative breeders as cooperative (but see *Methods*: *Variables*: *Re-classification)*.

*Re-classification*. Some classifications are ambiguous and multiple different classifications can be justified (e.g. see previous example regarding migrant status in jackdaws). We therefore tested whether re-classifying variables changed model results. We re-classified one ecological and one social variable. "Partial migrants", where at least one population of a species migrates but other populations are resident, were re-classified as residents. Facultative or suspected cooperative breeders were re-classified as non-cooperative breeders, rather than cooperative breeders.

*Developmental variables*. While many comparative studies include developmental mode and parental care as developmental covariates in their models, there was no variation in these variables within our Corvides dataset. Some studies include more detailed developmental variables such as incubation period and days until fledging (e.g. [54, 55]). We chose not to include these variables in our models because, due to limited data availability, this would have substantially reduced our sample size.

## Statistical modelling

All statistical analyses were undertaken in R v4.0.2 [70]. We used a phylogenetic generalized least squares (PGLS) modelling framework [71] in the package *caper* [72], which controls for non-independence of datapoints due to relatedness. We used this method because it is the most commonly used technique in the comparative brain size literature (e.g. [21, 36, 39, 42, 55, 60]). We constructed a consensus tree by downloading 1000 equally plausible phylogenetic trees for the species in our sample from www.BirdTree.org [21]. We used the Hackett rather than Ericson backbone because it is the most recently constructed; however, differences between backbones are small and they tend to produce consistent results [73]. Using TreeAnnotator in BEAST v1.10.4 [74], a maximum clade credibility consensus tree was built from

**Table 1. Qualitative conclusions drawn from models testing different hypotheses (SIH/EIH/combined).** Significant predictors are shown in bold in column 3.

| Hypothesis | Species number | Model predictors | Conclusions |
|---|---|---|---|
| **Ecological Intelligence Hypothesis (EIH)** | 59 | body size + diet breadth + environmental variation (*temperature variation*) + **movement** | 1. Species movement correlates with brain size (residents have bigger brains than nomads)<br>2. Environmental variation does not correlate with brain size |
| **EIH (PPC)** | 46 | body size + diet breadth + **environmental variation** (***PPC1*** + ***PPC2***) + movement | 1. Species movement *is not* correlated with brain size<br>3. Environmental variation *is* correlated with brain size: more environmental variation correlates with bigger brains |
| **Social Intelligence Hypothesis (SIH)** | 59 | body size + cooperative breeding + **social foraging** | 1. Social foraging correlates with brain size (species that forage in non-nested small groups have bigger brains than those that forage in pairs) |
| **Combined (EIH + SIH)** | 59 | body size + diet breadth + environmental variation (temperature variation) + **movement** + cooperative breeding + **social foraging** | 1. Species movement correlates with brain size (residents have bigger brains than nomads)<br>2. Social foraging correlates with brain size (species that forage in non-nested small groups *and solitarily* have bigger brains than those that forage in pairs) |

these equally plausible trees. This tree was then used to control for phylogenetic non-independence in the following PGLS models. Lambda was estimated using Maximum Likelihood. Model diagnostics and variance inflation factor (VIF) were checked to ensure assumptions were met and variables were not unacceptably collinear, respectively.

*Corvides analysis* Here, we tested how variable combination, source (environmental variability: temperature variation or PPC) and classification (partial migrant/resident; cooperative breeder/non-cooperative breeder) changed conclusions. See Table 1 for a summary of model formulations. We used Sayol et al.'s (2016) [21] brain size data only. We chose to use this dataset because only one method was used to measure brain size, and all body mass data came from the same specimens that brain volume was measured from. It is therefore likely to be the most precise data currently available. Using this data, we built three models: an SIH model (brain size in response to body size, cooperative breeding and social foraging), an EIH model (brain size in response to body size, migration, environmental variability and diet breadth), and a 'combined' model with all covariates included. For the EIH model, we tested two sources of environmental variation: temperature variation [39] and PPCs [21]. We used temperature variation as the measure of environmental variation in the combined model because the limited number of species with PPC data available resulted in some social foraging categories having extremely limited sample sizes. We tested 'initial' and 'reclassified' variables for species movement and cooperative breeding across all models. Brain and body size measurements were log-transformed.

## Results

### Aim 1—Quantifying variation in brain and body size between datasets

Altogether we collated brain size data for 1473 bird species. Of these, 1057 species had brain measurements in more than one dataset. Fig 1A visualises variation in log-transformed brain size estimates across datasets. All but one of the collated datasets had one brain size estimate per species; we therefore present García-Peña's [47] sex-separated data as sex-averaged in Fig 1A. Only two of the five studies that did not measure sexes separately explicitly stated that brain size datapoints were sex-averaged [18, 21]. While brain size estimates tended to be the mean value of multiple specimens (on average, ~six specimens per species), ten or more brain size estimates were based on a single speciman in at least two studies [46, 48].

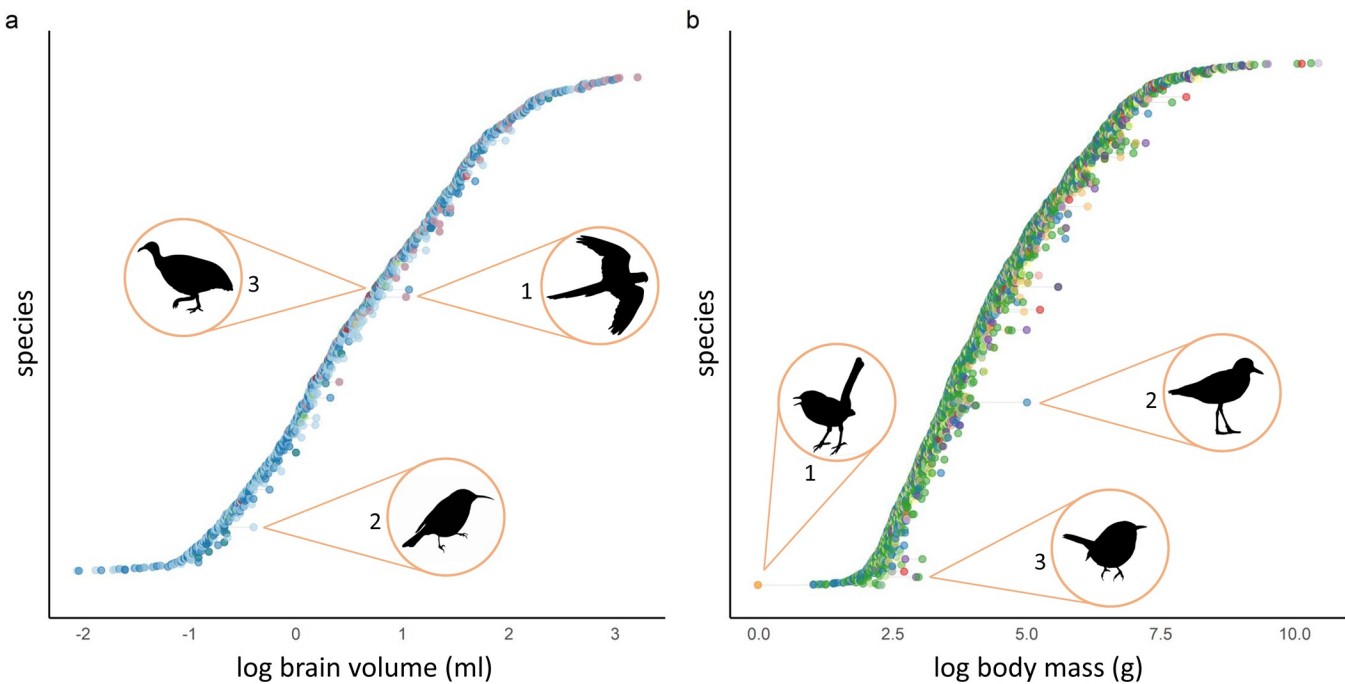

**Fig 1.**

Across datasets, some species varied considerably in brain size estimate (Fig 1A). For instance, the minimum and maximum brain size estimates of the maroon-bellied parakeet (*Pyrrhura frontalis*) (with the largest overall difference in estimates) overlapped with the brain size estimates of 159 other species in our full sample of 1473 species (thus, the brain size of this species spanned 10.79% of the species in our sample, depending on the dataset used). Similarly, the eastern spinebill's (*Acanthorhynchus tenuirostris*) minimum and maximum estimates (with the second largest difference) overlapped with 150 species (10.18% of our sample) and the great tinamou's (*Tinamus major*) estimates (the third largest difference) overlapped with 141 species (9.57% of our sample). Variation was not driven by any one dataset in particular (Fig 1A).

Our collated body size dataset contained 2331 species from eleven datasets. 1599 of these species had body mass measurements in more than one dataset. There was a general paucity of details for body mass data. Only one of 11 studies explicitly reported the number of samples from which body mass estimate was calculated [51]. Only two of 11 explicitly stated that body mass estimates were sex-averaged [53, 54]. Ten of the eleven datasets recorded both brain size and body mass data; however, only three of these *sometimes* collected body mass from the same specimens as brain size [18, 46, 55] and only two *always* collected body mass from the same specimens as brain size [21, 51]. Three studies did not mention the primary sources of the body mass data used [48, 52, 53].

Fig 1B visualises variation in log-transformed body mass estimates across datasets. Again, some species varied considerably in body mass estimates. For instance, the minimum and maximum body mass estimates of the blue-breasted fairywren (*Malurus pulcherrimus*), with the largest overall difference in estimates, overlapped with the body mass estimates of 140 other species in our full sample of 2331 species (6.01%). Similarly the three-banded plover's (*Charadrius tricollaris*) minimum and maximum estimates (with the second largest difference) overlapped with 759 species (32.56% of our sample), and the Carolina wren's (*Thryothorus*

*ludovicianus*) minimum and maximum estimates (with the third largest difference) overlapped with 429 species (18.36% of our sample). As in the brain size data, variation was not primarily driven by any one dataset (Fig 1B).

## Aim 2—Influence of variable inclusion, classification and source on model results

The full dataset used for this analysis included 59 species, where all species had known social and ecological variables (excluding PPCs). The PPC subset contained 46 species. Conclusions on the principal drivers of brain size evolution in Corvides differed depending on modelling approach (see Table 1 for a general summary; see Table 2 and Fig 2 for model results).

In the EIH model with temperature variation included, species movement was significantly associated with brain size. Specifically, resident species were found to have bigger brains than nomadic species. In the EIH model with PPCs (one measure of environmental variation) rather than temperature variation (a different measure of environmental variation), both PPC1 and PPC2 were significantly associated with brain size. However, species movement was not. Thus, depending on where we sourced our measure of environmental variation, we could

**Table 2. Phylogenetic generalised least squares model results, comparing different model formulations.** All significant pairwise contrasts for categorical variables are presented. Significant predictors are shown in bold.

| Model type | Predictors | λ | Estimate | SE | T-value | P-value |
|---|---|---|---|---|---|---|
| **EIH** | **Body size** | **0.625** | **0.663** | **0.03** | **22.434** | **<0.001** |
| | Diet breadth | | 0.184 | 0.377 | 0.487 | 0.629 |
| | Temperature variation | | -0.019 | 0.039 | -0.479 | 0.634 |
| | Movement (partial) | | 0.133 | 0.086 | 1.542 | 0.129 |
| | **Movement (resident)** | | **0.186** | **0.083** | **2.227** | **0.030** |
| | Movement (migrant) | | 0.185 | 0.115 | 1.611 | 0.113 |
| **EIH (PPC)** | **Body size** | **0.948** | **0.605** | **0.034** | **17.952** | **<0.001** |
| | Diet breadth | | 0.392 | 0.398 | 0.984 | 0.331 |
| | **PPC1** | | **0.013** | **0.006** | **2.207** | **0.033** |
| | **PPC2** | | **0.031** | **0.015** | **2.072** | **0.045** |
| | Movement (partial) | | 0.054 | 0.061 | 0.887 | 0.381 |
| | Movement (resident) | | -0.006 | 0.085 | -0.075 | 0.940 |
| **SIH** | **Body size** | **0.698** | **0.668** | **0.029** | **22.685** | **<0.001** |
| | Cooperative breeding (binary) | | -0.072 | 0.053 | -1.358 | 0.180 |
| | Social foraging (nested pairs) | | 0.079 | 0.061 | 1.292 | 0.202 |
| | Social foraging (solitary) | | 0.111 | 0.075 | 1.473 | 0.147 |
| | **Social foraging (non-nested small groups)** | | **0.153** | **0.068** | **2.238** | **0.030** |
| | Social foraging (nested small groups) | | 0.062 | 0.111 | 0.560 | 0.578 |
| **Combined (EIH + SIH)** | **Body size** | **0.662** | **0.658** | **0.031** | **21.565** | **<0.001** |
| | Diet breadth | | 0.161 | 0.389 | 0.414 | 0.680 |
| | Temperature variation | | 0.000 | 0.038 | -0.001 | 0.999 |
| | Movement (partial) | | 0.070 | 0.863 | 0.812 | 0.421 |
| | **Movement (resident)** | | **0.173** | **0.082** | **2.0967** | **0.041** |
| | Movement (migrant) | | -0.018 | 0.135 | -0.132 | 0.896 |
| | Cooperative breeding (binary) | | -0.079 | 0.055 | -1.430 | 0.159 |
| | Social foraging (nested pairs) | | 0.120 | 0.061 | 1.966 | 0.055 |
| | **Social foraging (solitary)** | | **0.226** | **0.099** | **2.291** | **0.027** |
| | **Social foraging (non-nested small groups)** | | **0.165** | **0.070** | **2.363** | **0.022** |
| | Social foraging (nested small groups) | | 0.112 | 0.114 | 0.981 | 0.332 |

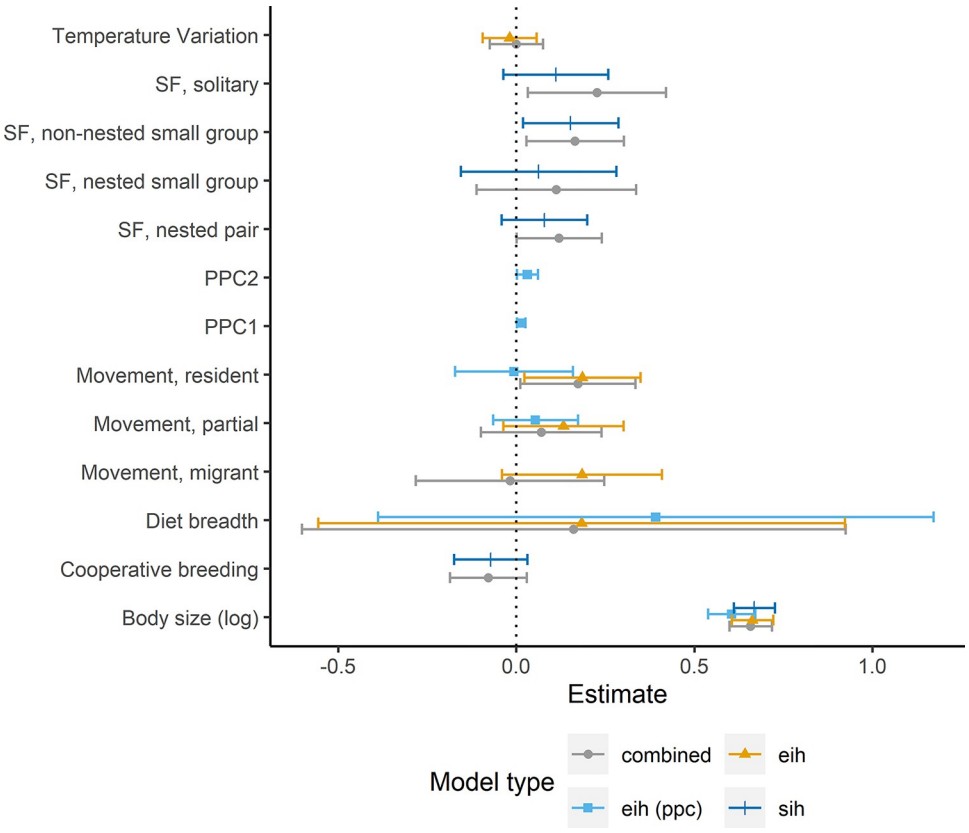

**Fig 2.**

have concluded that environmental variation drives brain size and species movement does not or, indeed, the exact opposite. Social variables also changed in significance depending on model specification. In the SIH model, species that forage in non-nested small groups were shown to have significantly larger brains than species that forage in pairs. In the combined model, with both social and ecological variables included, solitary foragers and species that forage in non-nested small groups were shown to have significantly larger brains than species that forage in pairs.

Changing facultative or suspected cooperative breeders from their initial categorisation of cooperative breeders to non-cooperative breeders did not qualitatively change SIH model results. However, changing partial migrants to residents did change EIH model results. For the EIH model with PPCs, PPC1 and PPC2 changed from significantly influencing brain size (PPC1: $\beta = 0.01$, SE = 0.01; 95% CI[0.001,0.03]; p = 0.03; PPC2: $\beta = 0.03$; SE = 0.02; 95% CI [0.002,0.06]; p = 0.04) to having no significant effect (PPC1: $\beta = 0.01$, SE = 0.01; 95% CI [0.0001,0.02]; p = 0.055; PPC2: $\beta = 0.03$, SE = 0.01; 95% CI[-0.002,0.06]; p = 0.07). In the combined model, where both changed variables were included, solitary foraging (relative to foraging in pairs) changed from having a significant ($\beta = 0.23$, SE = 0.10; 95%, CI[0.09,0.37]; p = 0.03) to no significant effect on brain size ($\beta = 0.16$, SE = 0.10; 95%, CI[0.03,0.35]; p = 0.11).

Note, however, that although results were unstable in regards to significance, and thus in qualitative conclusions, effect sizes and confidence intervals were not drastically different between models.

## Discussion

In agreement with a growing body of literature [29–32], our analyses raise concerns that comparative analyses of brain size studies may not provide robust means of testing hypotheses about cognitive evolution. We show that there is considerable variation in bird brain and body size estimates across datasets (Fig 1), most of which is likely to be due to intraspecific variation. The most common methods of comparative brain size analysis do not take this variation into account, although it has the potential to substantially influence results. The combination, source and classification of social and ecological variables also changed results (Fig 2). Indeed, we could have come to several contradictory conclusions depending on our modelling approach. Our results chime with and add to concerns raised in the primate comparative cognition literature [31, 32] that current methods in the comparative study of cognitive evolution, using brain size as a proxy of cognitive ability, give unreliable results.

Comparative brain size studies that test hypotheses of cognitive evolution typically use brain and body sizes averaged from multiple specimens of a single species to obtain one brain and body size estimate per species. Despite all brain sizes in our sample being estimated either from endocranial volume, or brain mass converted to volume (which has a strong positive correlation with volume [50]), we found substantial variation in brain size estimates across datasets. In several extreme cases, the minimum and maximum brain size measures for one species overlapped with 10% of our sample. Similarly, we found considerable variation between body size estimates of the same species: in extreme cases, the minimum and maximum body size of a species overlapped with more than 30% of the species in our sample.

Variation in brain size was not driven by any dataset in particular, suggesting that it was not the result of a specific methodological approach, and that it is likely to be the result of natural intraspecific variation. Indeed, previous work has found that brain size shows substantial intraspecific variation across taxa, including birds [75]. In some species of bird, sex and age class have been shown to be particularly important predictors of brain size [76]. Nevertheless, key information such as sex and age of specimens, and variation around the single reported estimate, were not reported for most brain size datasets. Body mass estimates had even sparser associated information: in most cases, the number of specimens from which the estimate was derived was not given, and neither was sex (an important covariate of body mass in many birds [77]) or variation around the estimate. The fact that brain and body size estimates tend to be derived from few (or an unknown number of) specimens, often of unrecorded sex and age, raises concerns as to how well they represent species-average values. Our results suggest that variation in brain and body size estimates across datasets is sometimes substantial. While the source of variation between datasets is likely to be due to intraspecific variation in the majority of cases, variation may sometimes be due to errors in the datasets. For instance, the variation between blue-breasted fairywren estimates is likely to be due to a typo in one dataset, where the species' body mass was recorded as 0.98 grams [18] but should have been an order of magnitude higher (other datasets record this species' body mass as 9.8 grams). At best, variation between datasets–whether due to natural intraspecific variation or due to errors in data entry–may introduce noise into comparative brain size studies, and at worst may lead to spurious conclusions about the drivers of cognitive evolution. We were not able to quantify the influence of variation between datasets on comparative brain size model results here, due to non-independence of the datasets, but this would be a valuable focus for future work. Another important and related issue is that recent research suggests the relationship between brain and body size is often taxa-dependent [78, 79]. Thus, the popular method of including body size as a covariate in models, in order to control for the relationship between brain and body size across a diverse range of species, may be flawed even when brain and body size estimates are accurate.

As well as investigating variation in brain and body size estimates across datasets, we used data for the Corvides infraorder to interrogate how robust models are depending on variable combination, source and classification. Researchers typically have varying approaches to model building, and in agreement with Wartel et al.'s (2019) [32] analyses of primate brain size data, we found that modelling approach influenced results. For instance, depending on whether we chose to include social variables or social and ecological variables, we could have concluded that there is no difference in brain size between species that forage in pairs and species that forage solitarily *or* that species that forage solitarily have significantly larger brains than those that forage in pairs. In addition, we showed that the source of covariates has the potential to substantially change results. Using temperature variation from Fristoe et al. (2017) [39] as a proxy of environmental variation resulted in no support that environmental variation drives the evolution of bigger brains. Meanwhile, using more detailed measures of environmental variation from Sayol et al. (2016) [21] resulted in support. Note, however, that models with temperature variation rather than PPC had a larger sample size, which may influence these results. Nevertheless, these findings parallel those reported in the primate brain size literature [31], where using differing variable sources resulted in differing results even when sample sizes were matched.

In addition to the previous issues, we also show that variable classification can drastically influence results. Classifications of variables are sometimes subjective; for instance, species with both cooperatively and non-cooperatively breeding populations could be classified as either. We therefore changed categorical variables that could justifiably be re-classified, and tested how this influenced results. Re-classifying suspected/facultative cooperative breeders as non-cooperative breeders did not change SIH model results; however, re-classifying partial migrants (i.e., where at least one population of a species migrate) as residents substantially changed EIH model results. While two measures of environmental variation (PPC1 and PPC2) were significantly associated with bigger brains before re-classification, there was no significant effect following re-classification. Thus, depending on the classification of a different model covariate, we could have concluded that environmental variation is associated with bigger brains or that there is no association. It must be considered, however, that while P-values crossed the threshold of significance, estimates and confidence intervals for predictor variables did not substantially change. With larger sample sizes, models may be less volatile. Nevertheless, many studies of brain size evolution use sample sizes in the same range as ours (e.g. [22, 25, 36, 80, 81]). The concerns raised here are thus pertinent to a wide range of studies.

While our study focused on highlighting problems with current methods of using brain size to test hypotheses of cognitive evolution, there are also conceptual issues with such studies. A key and often overlooked point is that the relationship between relative brain size and cognitive ability is not clear [29, 30, 38]. Thus, even if studies show a strong correlation between specific variables and brain size, it is difficult to draw strong conclusions about how this translates to selective forces acting on cognition itself. In addition to this rather fundamental issue, we add our voices to a growing number in the field who argue that framing the SIH and EIH as dichotomous and competing hypotheses is not logically sound. The hypothesised underlying driver of cognitive evolution for both hypotheses is variation in environmental conditions (including social environment) in which individuals must gather and process information to mitigate uncertainty [82, 83]. Thus, according to theory, the foundational driver of cognitive evolution is the same between both the SIH and EIH. Moreover, social and ecological variables are not independent, i.e., social species solve ecological problems in a social context, and sociality itself may evolve in response to ecological variables [84, 85]. We therefore suggest not only that our methodological approach to studying comparative brain size evolution needs to change, but also the conceptual framework itself. Rather than splitting often-correlated

variables into dichotomous and competing predictors, we could benefit from quantifying the environmental uncertainty animals face in specific contexts and examining how this may drive cognitive evolution.

When considering the accumulating literature on issues associated with current methods in the comparative study of cognitive evolution using brain size (here; [29–32]), we add our voices to a growing number in the field suggesting that we must (i) use caution when interpreting the results of studies using the same methods as presented here, and (ii) start to move away from these methods when interrogating hypotheses of cognitive evolution. It is important to note that the work done by previous researchers using the methods available at the time have helped us to form a deeper understanding of which variables may influence brain size evolution, and is key to guiding our path forwards in regards to what future research should endeavour to focus on. As more data become available and methods improve further, broad-scale comparative brain size studies will still offer novel and valuable insights into cognitive evolution [86]. However, we suggest that for a more nuanced understanding of the drivers of cognitive evolution we must also test how uncertainty in the social and ecological environment influences cognitive performance at the intra-specific level [84, 87–89], and between closely related species [90–94]. Conceptually, we also recommend a shift away from the treatment of the SIH and EIH as dichotomous and competing hypotheses. We propose that instead, we should work to understand whether and how uncertainty across a range of contexts drives cognitive evolution.

## Supporting information

**S1 Dataset. Body size data.**
(NEX)

**S2 Dataset. Brain size data.**
(CSV)

**S3 Dataset. The Corvides dataset as used in the analysis script (S1 Script).**
(CSV)

**S4 Dataset. The Corvides dataset including sources for each variable.**
(XLSX)

**S1 Data.**
(CSV)

**S1 Script. The script used for all analyses and figures.**
(RMD)

## Acknowledgments

Thank you to Matilda-Jane Brindle for help with creating Fig 1.

## Author Contributions

**Conceptualization:** Rebecca Hooper, Becky Brett, Alex Thornton.

**Data curation:** Rebecca Hooper, Becky Brett.

**Formal analysis:** Rebecca Hooper.

**Funding acquisition:** Rebecca Hooper.

**Investigation:** Rebecca Hooper, Becky Brett, Alex Thornton.

**Methodology:** Rebecca Hooper, Becky Brett, Alex Thornton.

**Project administration:** Rebecca Hooper, Becky Brett.

**Supervision:** Alex Thornton.

**Writing – original draft:** Rebecca Hooper.

**Writing – review & editing:** Rebecca Hooper, Alex Thornton.

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
