## [Decision Letter · Decision Letter 0]

16 Feb 2022

PONE-D-22-00405Problems with comparative analyses of avian brain sizePLOS ONE

Dear Dr. Hooper,

Thank you for submitting your manuscript to PLOS ONE. After careful consideration, we feel that it has merit but does not fully meet PLOS ONE’s publication criteria as it currently stands. Therefore, we invite you to submit a revised version of the manuscript that addresses the points raised during the review process.

We look forward to receiving your revised manuscript.

Kind regards,

Vitor Hugo Rodrigues Paiva, Ph.D.

Academic Editor

PLOS ONE

Journal Requirements:

"This work was funded by a Natural Environment Research Council GW4 studentship (grant no. NERC 107672G) to RH and a Leverhulme grant (grant no. RGP-2020-170) to AT. Resources from Thomas Currie’s PGLS workshop were used as a guide for the analysis."

"This work was funded by a Natural Environment Research Council GW4 studentship (grant no. NERC 107672G) (https://nerc.ukri.org/) to RH and a Leverhulme grant (grant no. RGP-2020-170) (https://www.leverhulme.ac.uk/) to AT. The funders had no role in study design, data collection and analysis, decisio"

Reviewers' comments:

Reviewer's Responses to Questions

**Comments to the Author**

1. Is the manuscript technically sound, and do the data support the conclusions?

Reviewer #1: Yes

Reviewer #2: Yes

Reviewer #3: Yes

Reviewer #4: No

2. Has the statistical analysis been performed appropriately and rigorously? 

Reviewer #1: Yes

Reviewer #2: Yes

Reviewer #3: Yes

Reviewer #4: No

3. Have the authors made all data underlying the findings in their manuscript fully available?

Reviewer #1: Yes

Reviewer #2: Yes

Reviewer #3: Yes

Reviewer #4: Yes

4. Is the manuscript presented in an intelligible fashion and written in standard English?

Reviewer #1: Yes

Reviewer #2: Yes

Reviewer #3: Yes

Reviewer #4: Yes

5. Review Comments to the Author

Reviewer #1: Original Submission:

1.1 Recommendation

Revise Major

2. Comments to Author

Ms. Ref. No.: PONE-D-22-00405

Title: Problems with comparative analyses of avian brain size

Overview and general recommendation:

Brilliant job! A very intriguing and eye-opening manuscript to the issues around the different analyses. I thoroughly enjoyed reading and reviewing it. The methodology was beautifully written and very detailed to allow a straightforward replication if required by others. The methodology appeared to suit the aims which was justified in the discussion section. The methodology had a diverse range of variables used to justify the aims which displays a lot of thought went into this study/manuscript. However, there were some sections which were a bit difficult to read. For example, the aims were jumbled around in the introduction rather than in a separate paragraph, so it was a bit tricky to tease out the aims. It would be useful if this was addressed. Otherwise, it was a great manuscript with some logical suggestions to improve this area of research.

2.1 Major comments:

• It was a bit difficult to tease apart the aims. One aim was mentioned in line 61 about “Here we interrogate the potential pitfalls…”. I got the idea that you were about to go into the method section immediately after that sentence as you just stated an aim. However, there was another paragraph about brain size and cognitive ability afterwards as it was still part of the introduction. There was another similar situation in line 73 with a sentence starting with “Here, we collate data from multiple datasets…”. I understand where you are coming from and why you wanted to place some aims there as it matches the paragraph topic, but maybe there should be a separate paragraph at the end of the introduction and before the methods which is solely on the aims. That way you can clearly define your aims for the reader. This is just an example of how you could write it:

“Aims:

Aim 1 – to interrogate the potential pitfalls of…

Aim 2 – to interrogate the brain estimates across datasets…”

This will help a lot with the flow and create an easier read for the readers.

I also notice you mention the aims again in lines 102 and 103 so maybe no need to mention it in line 73 and 61? Please readjust the aim section to make it more understandable.

• Line 93 to line 101 – this sounds very much like a methodology section, but it is located in the introduction. I suggest shifting this section to the methodology section and entitle it “study species” or something similar.

• The introduction into the topic of cognition and brain sizes was good. Maybe some more examples/references of the SIH and EIH in different animal groups could be used before primates are mentioned in line 51. This could give the readers a better understanding of the research around this topic and reinforce the idea that the two different cognitive comparative studies can produce different outcomes.

• Title – the title is great and suits the manuscript, but maybe you could add something about cognitive ability into it. This is since you reference it multiple times in the manuscript and if I am correct, are trying to see how robust the different methodologies are by testing for it using the brain size as a proxy.

• Line 252, 254 and 256 – just the species name was used such as Pyrrhura frontalis and Acanthorhynchus tenuirostris. Please use the common names as well since some readers may not know what animal you are referring to. You can add the species name beside the common name immediately after. For example, you could write line 252 with “…the minimum and maximum brain size estimates of the maroon-bellied parakeet, Pyrrhura frontalis, overlapped…” or use a similar method you used in line 201 and 202. It makes it a bit easier for readers who do not know what the common name of the Pyrrhura frontalis is so they know which bird you are talking about. This is considering I too had no idea what that bird or the others mentioned in line 254 and 256 were until I googled it.

2.2 Minor comments:

• Line 37 – in the abstract section, can you please specify/modify the phrase “direct measures” to explain what it is.

• Line 289 and 290 – these are good tables but, if possible, please fit one table each onto a single page? Maybe making the landscape orientation for the page specifically for the table horizontal? Then it can go back to being vertical for the writing section.

• Line 290 – in table 2, column λ, please make the numbers horizontal.

• Some of the paragraphs in the discussion section are very long. In line 322-343, please separate it into separate paragraphs. Also, the paragraph in lines 344-376 is extremely long. Please separate it into different chunks for a smoother read.

• Line 97 – the “Corvides infraorder” was mentioned. Please explain what a Corvide infraorder is so readers who do not know will know that it is.

Reviewer #2: What a pleasure was to read such an interesting work. After reading the text I have nothing but a few comments, most of whish has been directly added to the attached document.

My main concerns refer to the criteria you used o delimiting some of your variables, e.g., the variable “Movemets”, which is, at least as stated, an oversimplification of a complex phenomenon. This by no means invalidates your methods and the conclusion (at least this my general impression); however, I would like you elaborate more on those variables.

I also have had difficulties to follow the introductory section. Nothing serious indeed, but at the end of each paragraph you provide one of your objectives, which isadequtely summarized in the last. Therefore, I suggest you revise the text to provide the rationale of your study, including the well noticed limits of the current approaches in measuring the avian brain size, and theb lef the objctives to the last one.

In summary, I believe your text will benefit from a review of your text on the variables, epscially the criteria you used for delimiting the categories. Please, also review some sentences throughout the text (see the attached document) which need be supported by appropriate references.

Reviewer #3: A well-written and concise text that highlights some fundamental questions about how we study brain size in relation to the evolution of cognition. The authors compile a large number of measurements, from data sets that are often used in these types of studies, quantifying the differences between them and how they affect the results in different analyses. Their results show how the use of different datasets and classifiers changes the results and thus the support for SIH or EIH when Corvides is analyzed as a whole.

In general, the text is fine, and I think it is appropriate to focus on the broad implications of these results for the field. However, I want to offer some comments and suggestions to the authors that I think can improve the quality of this paper:

a) The authors do very little analysis of the entire data set, limiting themselves to indicating how disparate some measurements are. With such a large data set, one would assume that the study could run various analyzes similar to those done on Corvides, to show how widespread these problems are. However, the selection of this last group is well justified and their results are quite illustrative regarding the point they are trying to make.

b) The figures would benefit from the addition of illustrations. For example, in Figure 1a you could highlight some of the species with the most variation by pointing to them with an arrow and showing their silhouette. The same can be done with 1b. You can also use colors or symbols to separate different classifier categories in Fig. 2. This would add more visual interest to the figures and make them easier to read.

c) The main problem with the current text is in its discussion. Overall it is fine, but some conclusions seem to be disconnected from the arguments presented. For example, intraspecific variation is suggested to be a major source of measurement discrepancies, but little evidence is offered for this. Indeed, this is likely to be a very important factor, as is the use of average measurements, but this inference would benefit from clear examples provided by studies or a direct measure of variation by researchers. The authors also suggest that a dichotomy/antagonism approach to testing the current intelligence hypotheses may not be the best way to address this issue, particularly for supraspecific levels. This again may be correct, but it does not necessarily follow from the results of the present study. It could be argued that, in fact, the correct approach could be to include as many variables as possible and avoid the use of average values, and still obtain results that can be interpreted in favor of one or the other hypothesis. Therefore, being aware of intraspecific variation and differences between data sets does not necessarily imply that, at least in some cases, these hypotheses cannot be tested as mutually exclusive. I generally agree with the authors' suggestions and inferences, but they tend to feel disconnected from the content of the results and thus may benefit from further discussion.

d) The question of whether brain size is a good predictor of cognitive ability is firmly established in the introduction and dropped entirely afterwards. This bothers me, because it stayed in the back of my mind throughout the text, and is never discussed in the context of the results. I understand that it is beyond the scope of this document to resolve this issue, but it is such an important question that the lack of closure or consideration during the discussion distracted me from the topic of the document.

e) The title is too broad. Consider changing it to better reflect the specific scope of the text.

Reviewer #4: The manuscript is an exploration of avian brain size data sets and a focused analysis of two widely cited hypotheses in brain evolution: the ecological intelligence and social brain hypotheses. Although the authors are correct that there are a myriad of issues with brain size research and what can be inferred from it, the conclusions are not supported by their analyses. More importantly, the authors do not offer any alternatives or solutions and their own analyses contain the same flaws that they identify in other analyses. The authors also overlook several key issues and constraints that are germane to their study. I provide more detailed comments below.

1. The Introduction sets up what is largely a false dichotomy of hypotheses. There are certainly some researchers who actively promote the social brain hypothesis over the ecological intelligence (or cognitive buffer) hypothesis. However, I would argue that most researchers in this field agree that there is not one hypothesis that can explain brain size evolution in any clade. So pitting these two hypotheses against one another might apply to the primate literature, but less so to all other comparisons that are being made in modern studies. If anything, most recent research effort has been dedicated to examining allometry, changes in evolutionary rate, and identifying whether it is body or brain size (or both) that change to result in species variation in relative brain size. The premise of the current study is therefore problematic.

2. The authors discuss the importance of intraspecific variation a lot in their manuscript and I agree that this is likely an issue in many broad comparative studies, not just those focused on brain size. However, they do not actually analyze intraspecific variation. Analysing multiple datasets that overlap with their original source and then extrapolating variances from that does not reflect intraspecific variation. The authors do not demonstrate that simply looking at variation across datasets that largely provide only a single averaged data point can be used to estimate intraspecific variation. Further, recent studies that have examined intraspecific allometry are not cited by the authors.

3. The analyses across all bird species are superficial. The authors compared datasets to see how variable they are, but no attempt was made to address the broader evolutionary hypotheses they outline in the Introduction. I also did not really follow what the %difference estimates were referring to. This is not described sufficiently in the Methods and not explained in the Results. I was eventually able to glean the meaning from Figure 1, but I am not convinced that this is an effective measure of intraspecific variation when they already indicated that there is a lot of overlap in data sources across studies. A lot of emphasis is also placed on variation that is <10%. I am not aware of anyone studying allometry who assumes that there is 0% variation in any measurement within species and many birds can vary by more than 10% across seasons, ages, and sexes, so having some species that appear as moderate outliers does not appear to be sufficient evidence to dismiss a subfield of research. I would also encourage the authors to ensure that typos in datasets were accounted for and that species were correctly identified.

4. Related to my above point, the authors do not discuss the many issues that plague a simple measurement like body mass. First, obtaining body mass data from museum specimens is often not possible. The majority of specimens in many museum collections lack body mass data or the body mass data is skewed because the specimen was found dead (so body parts might be missing, rotted, dessicated, etc.). The complaint of not having body mass data specific to each specimen for which brain size data is collected is therefore not one that can be readily solved, unless one kills thousands of specimens. Second, body mass is a highly variable measurement. Body mass changes daily within an individual, between ages, sexes, and seasons, and across geographic locations in the form of clines or subspecies. Entering the “correct” body mass for a species is therefore an almost impossible task. Some references provide sex specific body masses that can be averaged or ranges that can be incorporated into comparative analyses, but the authors do not discuss these. Third, accurate body mass estimates for some species might not be attainable. Charadrius tricollaris is likely a case in point. This is not a well studied species and the number of museum specimens is probably quite low, so accurate body mass estimates for this species might not be possible without measuring them in the field. So, yes body mass can be an issue in comparative studies of relative brain size, but not for the reasons the authors discuss.

5. The analyses of the Corvides data are problematic for several reasons. First, the authors did not discuss cooperative breeding group size. The originators of the SBH emphasized the importance of group size to support their hypothesis, not simply the presence/absence of social behaviour. Second, the authors overlooked the potential for developmental traits to influence relative brain size within this clade. Incubation period, duration of parental care, clutch size, and other variables related to reproduction and development have all been associated with relative brain size too. If the goal is to cast doubt on cognitive explanations of variation in relative brain size, then it would stand to reason to include these other variables. Third, finding lack of support for a hypothesis in one clade does not negate the hypothesis. This is especially true for brain size studies: some variables are important in some clades and not others. In fact, the originator of the SBH have emphasized this over and over again to explain the lack of concordance between primate data and that of other mammalian clades (as well as birds).

6. I did not understand how the conclusions of the authors followed logically from their results. For example, what data was presented that indicates that “brain size studies are not methodologically robust”? This statement implies that brain size is not measured properly or that the statistics used are inherently flawed, neither of which are shown by the authors. As indicated above, the authors do not effectively show the magnitude of intraspecific variation or how that would cause “unreliable results”. In fact, that phrase itself is somewhat inflammatory given the lack of evidence provided by the authors. In addition, the authors emphasize that different modelling approaches caused different results, which is entirely expected and not novel. The fact that the inclusion of different variables and combination of those variables generates different results is a given for any comparative study and does not mean that the subject is unreliable or flawed. It probably means that some variables covary with one another in complex ways, which is to be expected for studies of brain size. The authors also include a lot of discussion about variables used to test the EIH. This was neither warranted nor necessary. Somewhat bizarrely, the authors finally get to discussing the two hypotheses to state that they should not be considered dichotomous, suggesting that they began the premise of this manuscript with a straw-man argument. Finally, the end of the Discussion is not what can be taken away from this study. The relationship between social and ecological environment and intraspecific level cognitive performance need not be the same relationship we see across species. Indeed, neuron numbers do not reflect individual performance within mammalian species, but do seem to explain species differences in cognitive performance. Unfortunately, due to the theoretical and analytical issues I have outlined above, I am not sure anything can be taken away from the analyses presented in its current form.

7. Overall, I was also not impressed with the literature cited. Many broad statements were made that cited a cherry-picked subset of papers, while ignoring more recent contributions to the field of research. For example, there are several papers by Jeroen Smaers that emphasize other aspects of brain size evolution, like evolutionary rate changes and allometry, that would have been appropriate to include as part of the Introduction or Discussion, but were not. Similarly, recent studies on intraspecific allometry and the degree of intraspecific variation were also not cited, despite intraspecific variation being the focus of this study. A broader and more careful treatment of the literature is needed.

8. Minor comments

a. Please refrain from using the term “type specimen” in this context. That term specifically refers to the specimen for which a species description/identification is based on.

b. A better figure legend is needed for Figure 1 as it is a bit unclear at first what “species” is referring to on the y-axis.

6. PLOS authors have the option to publish the peer review history of their article (what does this mean?). If published, this will include your full peer review and any attached files.

Reviewer #1: No

Reviewer #2: No

Reviewer #3: **Yes: **Martin Chavez-Hoffmeister

Reviewer #4: No

---

## [Author Response · Author response to Decision Letter 0]

8 Apr 2022

Reviewer #1

Brilliant job! A very intriguing and eye-opening manuscript to the issues around the different analyses. I thoroughly enjoyed reading and reviewing it. The methodology was beautifully written and very detailed to allow a straightforward replication if required by others. The methodology appeared to suit the aims which was justified in the discussion section. The methodology had a diverse range of variables used to justify the aims which displays a lot of thought went into this study/manuscript. However, there were some sections which were a bit difficult to read. For example, the aims were jumbled around in the introduction rather than in a separate paragraph, so it was a bit tricky to tease out the aims. It would be useful if this was addressed. Otherwise, it was a great manuscript with some logical suggestions to improve this area of research.

Thank you for your positive and constructive feedback! We have adjusted the manuscript according to your points (detailed below), and believe that the manuscript is much improved as a result.

2.1 Major comments:

• It was a bit difficult to tease apart the aims. One aim was mentioned in line 61 about “Here we interrogate the potential pitfalls…”. I got the idea that you were about to go into the method section immediately after that sentence as you just stated an aim. However, there was another paragraph about brain size and cognitive ability afterwards as it was still part of the introduction. There was another similar situation in line 73 with a sentence starting with “Here, we collate data from multiple datasets…”. I understand where you are coming from and why you wanted to place some aims there as it matches the paragraph topic, but maybe there should be a separate paragraph at the end of the introduction and before the methods which is solely on the aims. That way you can clearly define your aims for the reader. This is just an example of how you could write it:

“Aims:

Aim 1 – to interrogate the potential pitfalls of…

Aim 2 – to interrogate the brain estimates across datasets…”

This will help a lot with the flow and create an easier read for the readers. 

I also notice you mention the aims again in lines 102 and 103 so maybe no need to mention it in line 73 and 61? Please readjust the aim section to make it more understandable.

We have re-organised the introduction so that the Aims are stated in the final paragraph (beginning L92), before moving onto the methods section.

• Line 93 to line 101 – this sounds very much like a methodology section, but it is located in the introduction. I suggest shifting this section to the methodology section and entitle it “study species” or something similar. 

We have moved this section to Methods (L118).

• The introduction into the topic of cognition and brain sizes was good. Maybe some more examples/references of the SIH and EIH in different animal groups could be used before primates are mentioned in line 51. This could give the readers a better understanding of the research around this topic and reinforce the idea that the two different cognitive comparative studies can produce different outcomes. 

We have added references to non-avian, non-primate taxa in which the SIH and EIH have been investigated using brain size (L50)

• Title – the title is great and suits the manuscript, but maybe you could add something about cognitive ability into it. This is since you reference it multiple times in the manuscript and if I am correct, are trying to see how robust the different methodologies are by testing for it using the brain size as a proxy.

This is an excellent point, and we have adjusted the title to reflect that our study is specifically focused on using brain size to study cognitive evolution in birds (L1).

• Line 252, 254 and 256 – just the species name was used such as Pyrrhura frontalis and Acanthorhynchus tenuirostris. Please use the common names as well since some readers may not know what animal you are referring to. You can add the species name beside the common name immediately after. For example, you could write line 252 with “…the minimum and maximum brain size estimates of the maroon-bellied parakeet, Pyrrhura frontalis, overlapped…” or use a similar method you used in line 201 and 202. It makes it a bit easier for readers who do not know what the common name of the Pyrrhura frontalis is so they know which bird you are talking about. This is considering I too had no idea what that bird or the others mentioned in line 254 and 256 were until I googled it.

Thank you – we have addressed this in L266, L269, L271, L285, L288, L290.

2.2 Minor comments:

• Line 37 – in the abstract section, can you please specify/modify the phrase “direct measures” to explain what it is. 

We have modified this phrase for clarity (L37 – L39)

• Line 289 and 290 – these are good tables but, if possible, please fit one table each onto a single page? Maybe making the landscape orientation for the page specifically for the table horizontal? Then it can go back to being vertical for the writing section. 

We have ensured that all tables fit onto one page (p.15, p.16)

• Line 290 – in table 2, column λ, please make the numbers horizontal. 

We have done this (p.16)

• Some of the paragraphs in the discussion section are very long. In line 322-343, please separate it into separate paragraphs. Also, the paragraph in lines 344-376 is extremely long. Please separate it into 

different chunks for a smoother read. 

We have shortened the sentences and split paragraphs in the discussion for ease of reading.

• Line 97 – the “Corvides infraorder” was mentioned. Please explain what a Corvide infraorder is so readers who do not know will know that it is. 

We have explained what the Corvides infraorder is (paragraph beginning L118).

Reviewer #2

What a pleasure was to read such an interesting work. After reading the text I have nothing but a few comments, most of whish has been directly added to the attached document.

We are so glad you enjoyed reviewing this manuscript. Thank you for your feedback, which has greatly improved the manuscript. Below we address all points raised.

My main concerns refer to the criteria you used o delimiting some of your variables, e.g., the variable “Movemets”, which is, at least as stated, an oversimplification of a complex phenomenon. This by no means invalidates your methods and the conclusion (at least this my general impression); however, I would like you elaborate more on those variables.

We have elaborated on how and why we decided to categorise variables as we did in the paragraph beginning L145.

I also have had difficulties to follow the introductory section. Nothing serious indeed, but at the end of each paragraph you provide one of your objectives, which is adequtely summarized in the last. Therefore, I suggest you revise the text to provide the rationale of your study, including the well noticed limits of the current approaches in measuring the avian brain size, and theb lef the objctives to the last one.

We have edited the introduction so that the aims are clearly stated in the final paragraph (beginning L92).

In summary, I believe your text will benefit from a review of your text on the variables, epscially the criteria you used for delimiting the categories. Please, also review some sentences throughout the text (see the attached document) which need be supported by appropriate references.

We have also integrated the comments you left on our manuscript; below are a list of these edits numbered by comment order:

1. We reviewed references to ensure they fit the style of the journal

2. We have added references to support our point that ‘most studies’ use a single measurement of brain size per species (L68)

3. We have added a reference to support this point (L70). 

5. Original manuscript L77 has been revised as requested (now L76)

6. We have added an example and references to support our point that variable classifications can be somewhat arbitrary (L86 – L91; paragraph beginning L211)

7. We have explicitly stated body mass rather than size, as requested (L112)

8. We have added a reference for the taxonomy we follow (see paragraph beginning L118)

9. We have added a reference to our Supplementary Material here (L124)

10. We have expanded on our definition of partial migrant, as requested (paragraph beginning L145)

11. We have added a reference to support our point that some species are only migratory in specific regions (L89)

12. We have added references to support out claim that social foraging and cooperative breeding have been considered potentially important drivers of cognitive evolution in the past (see paragraph beginning L169)

13. We have added references to support our classifications of group size (L189)

14. We have deleted the Methods section that explains the predictions of the SIH, given that this does not belong in the Methods

15. We have added references to support that variable classifications can be ambiguous (L86 – L91; paragraph beginning L211) 

16. We use both English and Latin names throughout following your suggestion for consistency

17. We clarify that we mean specifically in the comparative brain size literature, not the literature more generally (L113, L355)

Reviewer #3

A well-written and concise text that highlights some fundamental questions about how we study brain size in relation to the evolution of cognition. The authors compile a large number of measurements, from data sets that are often used in these types of studies, quantifying the differences between them and how they affect the results in different analyses. Their results show how the use of different datasets and classifiers changes the results and thus the support for SIH or EIH when Corvides is analyzed as a whole.

Thank you for your extensive and helpful review; we have responded to each of your points and believe that your feedback has greatly improved the manuscript.

In general, the text is fine, and I think it is appropriate to focus on the broad implications of these results for the field. However, I want to offer some comments and suggestions to the authors that I think can improve the quality of this paper:

a) The authors do very little analysis of the entire data set, limiting themselves to indicating how disparate some measurements are. With such a large data set, one would assume that the study could run various analyzes similar to those done on Corvides, to show how widespread these problems are. However, the selection of this last group is well justified and their results are quite illustrative regarding the point they are trying to make.

Thank you for this point. Given our emphasis on finding fine-scale, accurate social and ecological data for each species – including justified reclassifications of variables – we were limited in the number of species we could use for the model instability analysis. We hope that the justifications in the paragraph beginning L238 clarify why the Corvides represent an excellent clade in which to run these analyses. Due to limitations with the brain and body size data available for the Corvides (brain/body size estimates are often identical between datasets, down to decimal point accuracy, indicating the datasets are not fully independent), we could not perform this analysis for the Corvides alone (see L124-L127).

b) The figures would benefit from the addition of illustrations. For example, in Figure 1a you could highlight some of the species with the most variation by pointing to them with an arrow and showing their silhouette. The same can be done with 1b. You can also use colors or symbols to separate different classifier categories in Fig. 2. This would add more visual interest to the figures and make them easier to read.

This is an excellent idea. Fig. 1 has been edited to include silhouettes of species with the most variation in brain/body size, respectively (p. 12). Fig. 2 has been edited so each dataset is also represented by a different symbol (p. 17).

c) The main problem with the current text is in its discussion. Overall it is fine, but some conclusions seem to be disconnected from the arguments presented. For example, intraspecific variation is suggested to be a major source of measurement discrepancies, but little evidence is offered for this. Indeed, this is likely to be a very important factor, as is the use of average measurements, but this inference would benefit from clear examples provided by studies or a direct measure of variation by researchers.

Thank you for this point. Unfortunately, because the brain and body size datasets used often share datapoints with older datasets (e.g. newer datasets often build off the back of older ones, and it is sometimes unclear if datapoints are indeed shared or replicated), it was not possible for us to robustly quantify the influence that different datasets have on the conclusions of studies in the whole data or the Corvides dataset. This is now explained in the text (L124-L127). We have expanded our section on intraspecific variation in the discussion to support our argument that this is an important factor to consider (see paragraph beginning at L349).

The authors also suggest that a dichotomy/antagonism approach to testing the current intelligence hypotheses may not be the best way to address this issue, particularly for supraspecific levels. This again may be correct, but it does not necessarily follow from the results of the present study. It could be argued that, in fact, the correct approach could be to include as many variables as possible and avoid the use of average values, and still obtain results that can be interpreted in favor of one or the other hypothesis. Therefore, being aware of intraspecific variation and differences between data sets does not necessarily imply that, at least in some cases, these hypotheses cannot be tested as mutually exclusive. I generally agree with the authors' suggestions and inferences, but they tend to feel disconnected from the content of the results and thus may benefit from further discussion.

Thank you for pointing out the disconnect between these ideas. We have re-organised the discussion so that these points follow a more logical order.

d) The question of whether brain size is a good predictor of cognitive ability is firmly established in the introduction and dropped entirely afterwards. This bothers me, because it stayed in the back of my mind throughout the text, and is never discussed in the context of the results. I understand that it is beyond the scope of this document to resolve this issue, but it is such an important question that the lack of closure or consideration during the discussion distracted me from the topic of the document.

Thank you for this point. We have elaborated on this in the discussion (see paragraph beginning at L412, specifically L413 – L417).

e) The title is too broad. Consider changing it to better reflect the specific scope of the text.

We agree that the title was too broad, and have changed the title to better reflect the topic of the study (L1).

Reviewer #4

The manuscript is an exploration of avian brain size data sets and a focused analysis of two widely cited hypotheses in brain evolution: the ecological intelligence and social brain hypotheses. Although the authors are correct that there are a myriad of issues with brain size research and what can be inferred from it, the conclusions are not supported by their analyses. More importantly, the authors do not offer any alternatives or solutions and their own analyses contain the same flaws that they identify in other analyses. The authors also overlook several key issues and constraints that are germane to their study. I provide more detailed comments below.

Thank you for your review of our manuscript. You have raised some interesting points, many of which we feel have improved our manuscript (see below). However, we disagree that the conclusions of our paper are not supported by the analyses (see comments below). We are not proposing that our analyses offer the solution to the issues raised. Rather, we use the same analyses as previous studies and highlight the issues that they have. We then move on to discuss alternative methods that will enable researchers to gain a deeper insight into cognitive evolution. We acknowledge that, given the contentious nature of the issues we discuss, there are likely to be important disagreements. Indeed, the other three reviewers were largely positive about the manuscript, with reviewers 1 and 2 in being particularly complementary. It would be naïve of us to hope to persuade everyone, but we hope that our revisions to the manuscript will allow readers to evaluate our findings and arguments on their merits.

1. The Introduction sets up what is largely a false dichotomy of hypotheses. There are certainly some researchers who actively promote the social brain hypothesis over the ecological intelligence (or cognitive buffer) hypothesis. However, I would argue that most researchers in this field agree that there is not one hypothesis that can explain brain size evolution in any clade. So pitting these two hypotheses against one another might apply to the primate literature, but less so to all other comparisons that are being made in modern studies. 

We disagree that we set up a false dichotomy. It is common to see the EIH and the SIH treated as dichotomous explanations of brain size evolution in comparative cognition papers. To give some recent examples across a range of taxa: 

“Ecological hypotheses mainly involve investigating the cognitive demands associated with foraging … In contrast to ecological hypotheses, the social brain hypothesis (SBH) suggests sociality − specifically the cognitive demands of tracking, negotiating and maintaining social relationships − to be the main driving force behind variation in primate brain sizes … In carnivores, evidence suggests ecological variables … are influencing brain size; whereas, no support is found for the social brain hypothesis” (Chambers et al., 2021, carnivores)

“Three explanatory frameworks—social, ecological and cognitive—roughly summarize different schools of thought about brain size evolution [6–12] … The debate about which of the three hypotheses best explains brain size evolution coincides with controversy over what specific variables select for the evolution of larger brains” (Todorov et al., 2021, marsupials)

“This idea is formally developed in the ‘cognitive buffer’ hypothesis (CBH, hereafter), which postulates that large brains evolved to facilitate behavioural adjustments to enhance survival under changing conditions … As predicted by the CBH … species that live in more variable environments also tend to have larger brains regardless of the type of variation” and “although according to the social intelligence hypothesis the demands of social living might have selected for enlarged brains, including factors that represent social behaviour … does not alter the patterns we report in the present study” (Sayol et al., 2016, birds)

“Contrary to the social brain hypothesis, new work suggests that ecological factors, rather than social complexity, best predict relative brain size across primate species.” (DeCasien et al., 2022, primates)

“Here, we use a much larger sample of primates, more recent phylogenies, and updated statistical techniques, to show that brain size is predicted by diet, rather than multiple measures of sociality, after controlling for body size and phylogeny... Our results call into question the cur- rent emphasis on social rather than ecological explanations for the evolution of large brains” (DeCasien et al., 2017, primates)

“social and ecological explanations for the emergence of complex cognition are often treated as rival hypotheses” (Rosati, 2017, primates)

If anything, most recent research effort has been dedicated to examining allometry, changes in evolutionary rate, and identifying whether it is body or brain size (or both) that change to result in species variation in relative brain size. The premise of the current study is therefore problematic.

We agree that a lot of research has recently been conducted into brain-body allometry. We cite this research in our manuscript (L374). However, there has been a plethora of recent research that examines the SIH and EIH in the way we discuss in our manuscript (e.g. see examples in the response to your comment above, all of which are papers published between 2016 and 2022).

2. The authors discuss the importance of intraspecific variation a lot in their manuscript and I agree that this is likely an issue in many broad comparative studies, not just those focused on brain size. However, they do not actually analyze intraspecific variation. Analysing multiple datasets that overlap with their original source and then extrapolating variances from that does not reflect intraspecific variation. The authors do not demonstrate that simply looking at variation across datasets that largely provide only a single averaged data point can be used to estimate intraspecific variation. Further, recent studies that have examined intraspecific allometry are not cited by the authors.

Thank you for this point. We realise that our discussion of intraspecific variation may not have been clear; indeed, we did not mean to communicate that we could directly measure intraspecific variation using the datasets in our study, merely that the variation observed is likely to – at least in part – be driven by intraspecific variation. Intraspecific variation is likely to influence conclusions of comparative brain size models; however, we were not able to robustly test the influence of variation between datasets because datasets often used overlapping sources and thus were not independent. We have included a line in the discussion to suggest that quantifying exactly how intraspecific variation might influence the results of comparative brain studies would be a fruitful line of future research (L372). We have also explained why we could not analyse the influence of intra-specific variation using the Corvides dataset (L124-L127). Furthermore, we have expanded our section on intraspecific variation in the discussion to support our argument that this is an important factor to consider (see paragraph beginning at L349), and have edited our wording throughout so that it is clear that we do not directly measure intra-specific variation (e.g. L93, L103, L363).

3. The analyses across all bird species are superficial. The authors compared datasets to see how variable they are, but no attempt was made to address the broader evolutionary hypotheses they outline in the Introduction. I also did not really follow what the %difference estimates were referring to. This is not described sufficiently in the Methods and not explained in the Results. I was eventually able to glean the meaning from Figure 1, but I am not convinced that this is an effective measure of intraspecific variation when they already indicated that there is a lot of overlap in data sources across studies. 

Thank you for this point. Given our emphasis on finding fine-scale, accurate social and ecological data for each species to address Aim 2 (how differing modelling decisions influence results), we were limited in the number of species we could use for the model instability analysis. Such detailed data is unavailable for a large majority of species. We hope that the justifications in the paragraph beginning L118 clarify why the Corvides represent an excellent clade in which to run these analyses. We could not use the Corvides dataset to robustly test the influence of variable brain/body size estimates between datasets due to non-independence of the available data (L124-L127). We have also clarified at various points throughout the manuscript that while the variation observed between datasets is likely to be the result of intraspecific variation, this is not known with certainty (see paragraph beginning L349), which hopefully alleviates your worry that we attempt to use % overlap to quantify intraspecific variation. We also note that the reviewer does not suggest an alternative route through which to quantify intra-specific variation.

A lot of emphasis is also placed on variation that is <10%. I am not aware of anyone studying allometry who assumes that there is 0% variation in any measurement within species and many birds can vary by more than 10% across seasons, ages, and sexes, so having some species that appear as moderate outliers does not appear to be sufficient evidence to dismiss a subfield of research. 

We did not mean to imply that we, or anybody, should expect 0% variation in brain or body size within species. Instead, we mean to highlight that using one mean datapoint per species (which most comparative brain studies do) is likely to be problematic given that measurements can vary by a large degree depending on the dataset used. A species that has a relatively small brain according to one dataset may have a considerably larger brain according to another, and the same is true for body size. As an example, we found that in one dataset the maroon-bellied parakeet was reported to have a brain size of 1.99cm3, while in another it was reported to have a brain size of 2.81cm3. This means that, depending on the dataset used, this species overlaps with 150 other species in our sample. Depending on the analysis being run, this at best adds noise to analyses, and at worst may lead to incorrect conclusions. 

I would also encourage the authors to ensure that typos in datasets were accounted for and that species were correctly identified.

After combing through our combined datasets we did indeed find species that had been recorded under synonymous Latin names between different datasets, and some instances of erroneous duplications. 

Thank you for highlighting that this may be an issue, this was extremely valuable and we’re very pleased to have caught these errors at the revision stage.

Below are synonymous Latin names we identified in (1) the brain size dataset (n = 11) and (2) the body size dataset (n = 68). We have adjusted the text to reflect the correct sample sizes in paragraphs beginning at L264 and L283. Figure 1 has been updated with the corrected data.

Brain size dataset

1. Agapornis personatus was recorded as both Agapornis personata and Agapornis personatus

2. Aratinga finschi was recorded as both Aratinga finschii and Aratinga finschi

3. Cacatua roseicapilla was recorded as both Cacatua roseicapilla and Cacatua roseicapillus

4. Campephilus guatamalensis was duplicated

5. Catharacta maccormicki was recorded as both Catharacta maccormickii and Catharacta maccormicki

6. Chionis alba was recorded as both Chionis albus and Chionis alba

7. Chlidonias hybridus was recorded as both Chlidonias hybrida and Chlidonias hybridus

8. Cyclopsitta diophthalma was recorded as both Cyclopsitta diophthalma and Cyclopsitta diophthalmica

9. Lagopus muta was recorded as both Lagopus muta and Lagopus mutus

10. Loriculus philippensis was recorded as both Loriculus philippensis and Loriculus philipensis

11. Polytelis alexandrae was recorded as both Polytelis alexandrae and Polytelis aleexandrae

Body size dataset

1. Acanthorhynchus tenuirostris was entered as both Acanthorhynchus tenuirostris and Acanthorynchus tenuirostris

2. Calyptorhynchus lathami was entered as both Calyptorhynchus lathamii and Calyptorhynchus lathami

3. Agapornis personata was entered as both Agapornis personata and Agapornis personatus

4. Agapornis pullaria was entered as both Agapornis pullaria and Agapornis pullarius

5. Anodorhynchus hyacinthinus was entered as both Anodorhynchus hyacinthinus and Anodorhynchus hyacinthus

6. Aphelocephala nigricincta was entered as both Aphelocephala nigrocincta and Aphelocephala nigricincta

7. Aphelocoma caerulescens was entered as both Aphelocoma coerulescens and Aphelocoma caerulescens

8. Ara ambiguus was entered as both Ara ambiguus and Ara ambigua

9. Ara chloropterus was entered as both Ara chloropterus and Ara chloroptera

10. Ara severus was entered as both Ara severus and Ara severa

11. Aratinga finschi was entered as both Aratinga finschi and Aratinga finschii

12. Artamus cyanopterus was entered as both Artamus cyanopterus and Artamus cyanoptera

13. Artamus leucorhynchus was entered as both Artamus leucorhynchus and Artamus leucorynchus

14. Brotogeris chrysopterus was entered as both Brotogeris chrysopterus and Brotogeris chrysoptera

15. Cacatua roseicapilla was entered as both Cacatua roseicapillus and Cacatua roseicapilla

16. Campephilus guatemalensis was entered as both Campephilus guatemalensis and Campephilus guatamalensis

17. Campylorhynchus brunneicapillus was entered as both Campylorhynchus bruneicapillus and Campylorhynchus brunneicapillus

18. Certhiaxis cinnamomeus was entered as both Certhiaxis cinnamomeus and Certhiaxis cinnamomea

19. Chalcophaps indica was entered as both Chalcophaps indica and Chalcophaps indiaca

20. Cissopis leverianus was entered as both Cissopis leverianus and Cissopis leveriana

21. Cisticola cherinus was entered as both Cisticola cherinus and Cisticola cherina

22. Climacteris rufus was entered as both Climacteris rufus and Climacteris rufa

23. Copsychus sauIaris was duplicated

24. Corcorax melanorhamphos was entered as both Corcorax melanorhamphos and Corcorax melanorhamphus

25. Cormobates leucophaea was entered as both Cormobates leucophaea and Cormobates leucophaus

26. Corvus cryptoleucus was entered as both Corvus cryptoleucus and Corvus cryptoleucos

27. Cyclopsitta diophthalma was entered as both Cyclopsitta diophthalma and Cyclopsitta diophthalmica

28. Cylarhis gujanensis was entered as both Cylarhis gujanensis and Cycylarhis gujanensis

29. Cyphorhinus aradus was entered as both Cyphorhinus aradus and Cyphorhinus arada

30. Delichon urbicum was entered as both Delichon urbicum and Delichon urbica

31. Dicrurus hottentottus was entered as both Dicrurus hottentottus and Dicrurus hottentotus

32. Donacobius atricapilla was entered as both Donacobius atricapilla and Donacobius atricapillus

33. Dysithamnus mentalis was entered as both Dysithamnus mentalis and Dystithamnus mentalis

34. Emberiza rutila was entered as both Emberiza ruttila and Emberiza rutila

35. Falcunculus frontatus was entered as both Falculuncus frontatus and Falcunculus frontatus

36. Glyphorynchus spirurus was entered as both Glyphorhynchus spirurus and Glyphorynchus spirurus

37. Helmitheros vermivorum was entered as both Helmitheros vermivorum and Helmitheros vermivorus

38. Hylocichla mustelina was entered as both Hylocichla mustelinus and Hylocichla mustelina

39. Hylophylax naevius was entered as both Hylophylax naevia and Hylophylax naevius

40. Lagopus muta was entered as both Lagopus muta and Lagopus mutus

41. Lonchura malacca was entered as both Lonchura malacca and Lonchura molacca

42. Loriculus phillippensis was entered as both Loriculus phillippensis and Loriculus phillipensis

43. Melanoptila glabrirostris was entered as both Melanoptila glabirostris and Melanoptila glabrirostris

44. Mionectes oleagineus was entered as both Mionectes oleagineus and Mionectes oligeneus

45. Muscisaxicola alpinus was entered as both Muscisaxicola alpina and Muscisaxicola alpinus

46. Phaenicophilus palmarum was entered as both Phaenicophilus palmarum and Phaenocophilus palmarum

47. Phalaropus fulicaria was entered as both Phalaropus fulicaria and Phalaropus fulicarius

48. Phylloscopus sibilatrix was entered as both Phylloscopus sibilatrix and Phylloscopus sibiliatrix

49. Pionites melanocephalus was entered as both Pionites melanocephala and Pionites melanocephalus

50. Platycercus elegans was entered as both Platycercus elegans and Platycercus elegans elegans

51. Platyrinchus cancrominus was entered as both Platyrinchus cancrominus and Platyrinchus concrominus

52. Polytelis alexandrae was entered as both Polytelis alexandrae and Polytelis aleexandrae

53. Prinia leucopogon was entered as both PriniaSchistolais leucopogon and Prinia leucopogon

54. Prionochilus plateni was entered as both Prionochilus plateni and Prionochilus plateri

55. Pyrenestes sanguineus was entered as both Pyrenestes sanguineus and Pyrenestes sanguineous

56. Rupicola peruvianus was entered as both Rupicola peruvianus and Rupicola peruviana

57. Saxicola torquatus was entered as both Saxicola torquatus and Saxicola torquata

58. Schiffornis turdina was entered as both Schiffornis turdina and Schiffornis turdinus

59. Sericornis magnirostra was entered as both Sericornis magnirostra and Sericornis magnirostris

60. Strigops habroptilus was entered as both Strigops habroptilus and Strigops habroptila

61. Tachyphonus delattrii was entered as both Tachyphonus delatrii and Tachyphonus delattrii

62. Tetrao urogallus was entered as both Tetrao urogallus and Tetrao urogallis

63. Tiaris olivaceus was entered as both Tiaris olivaceus and Tiaris olivacea

64. Todiramphus sanctus was entered as both Todiramphus sanctus and Todirhamphus sanctus

65. Touit delictissima was entered as both Touit delictissima and Touit delictissimus

66. Toxorhamphus iliolophum was entered as both Toxorhamphus iliolophum and Toxorhamphus iliolophus

67. Veniliornis passerines was entered as both Veneliornis passerines and Veniliornis passerines

68. Tyrannus savana was entered as both Tyrannus savana and Tyrannus savanna

4. Related to my above point, the authors do not discuss the many issues that plague a simple measurement like body mass. First, obtaining body mass data from museum specimens is often not possible. The majority of specimens in many museum collections lack body mass data or the body mass data is skewed because the specimen was found dead (so body parts might be missing, rotted, dessicated, etc.). The complaint of not having body mass data specific to each specimen for which brain size data is collected is therefore not one that can be readily solved, unless one kills thousands of specimens. Second, body mass is a highly variable measurement. Body mass changes daily within an individual, between ages, sexes, and seasons, and across geographic locations in the form of clines or subspecies. Entering the “correct” body mass for a species is therefore an almost impossible task.

Simply because one single ‘correct’ measurement is logically impossible to collect for any one species, does not mean that we cannot criticise the way body mass data is currently collected and reported. Using a single sex-averaged, age-averaged body mass datapoint averaged across an often-unreported number of individuals can and should be improved upon if these methods are to continue being used. The difficulty in getting accurate body mass data for some species adds weight to our argument that perhaps we should move away from methods that require these measurements. Directly measuring cognitive performance within and between closely related species allows us to interrogate cognitive evolution without the noise introduced by covariates that are difficult to accurately measure.

Some references provide sex specific body masses that can be averaged or ranges that can be incorporated into comparative analyses, but the authors do not discuss these.

Our criticism is that most comparative brain size studies do not include multiple measurements of brain or body size per species. We did discuss and included citations for the studies we used in our analyses that do incorporate multiple measurements (L258-L260).

Third, accurate body mass estimates for some species might not be attainable. Charadrius tricollaris is likely a case in point. This is not a well studied species and the number of museum specimens is probably quite low, so accurate body mass estimates for this species might not be possible without measuring them in the field. So, yes body mass can be an issue in comparative studies of relative brain size, but not for the reasons the authors discuss.

Data on the range of body masses are available for many species. However, we do agree that a lot of species do not have detailed body mass data reported. We believe, though, that this only adds weight to our argument. The inability to get accurate body mass data for some species is another reason for us to move away from methods that require these measurements.

5. The analyses of the Corvides data are problematic for several reasons. First, the authors did not discuss cooperative breeding group size. The originators of the SBH emphasized the importance of group size to support their hypothesis, not simply the presence/absence of social behaviour

We agree that this would have been interesting to test, but given that we only had 17 species that were cooperative breeders in our dataset, we did not have the power to test whether group size modulated the relationship between brain size and cooperative breeding. Moreover, we would like to note that it has also been argued that cooperative breeding generates selection on bigger brains relative to non-cooperative breeding, regardless of group size (Burkart et al., 2009). Thus, the way we ran our analyses does address predictions made in previous research, and utilises the same methods to test these predictions (e.g. Iwaniuk & Arnold, 2004 also use a model with a binary response variable of cooperative/non-cooperative breeding). Because we were principally interested in testing how robust model results were when using the same methods as previous researchers, rather than drawing novel conclusions about the drivers of cognitive evolution, we do not think that this approach is problematic. 

Second, the authors overlooked the potential for developmental traits to influence relative brain size within this clade. Incubation period, duration of parental care, clutch size, and other variables related to reproduction and development have all been associated with relative brain size too. If the goal is to cast doubt on cognitive explanations of variation in relative brain size, then it would stand to reason to include these other variables.

Thank you for this point. We did not include developmental variables for two reasons. The primary reason we did not include developmental variables is that the majority of comparative bird brain studies that examine drivers of cognitive evolution either do not include developmental variables, or include developmental mode (altricial/precocial) and/or parental care as the only developmental variables (e.g.(Beauchamp & Fernández-Juricic, 2004; Emery et al., 2007; Iwaniuk & Arnold, 2004; Overington et al., 2009; Pravosudov et al., 2007; Shultz & Dunbar, 2010; West, 2014). As our models use only the Corvides infraorder, there is no variation in developmental mode or parental care. All species in our dataset with known developmental mode/parental care have altricial development with extended biparental care. There was therefore no need to control for these variables in our models. Several comparative studies of bird brain size do, however, include more detailed developmental variables, such as incubation period and days until fledging (Minias & Podlaszczuk, 2017; Sayol et al., 2016; Sol et al., 2010). While we did initially collect incubation period and days until fledging, both of which have some variation across species in our dataset, data are not known for 10 species (incubation) and 13 species (fledging) of 59 species (TempVar dataset) or 9 (incubation) and 12 (fledging) species of 46 species (PPC dataset). Including these variables would thus have resulted in considerably less power in our analyses, by reducing sample size ~20 – 25%. For these reasons, we decided not to include developmental variables. Given that we are not drawing conclusions from our results, but merely using the same methods as others in the field to highlight some flaws in this type of analysis, we do not think that omitting these variables undermines our conclusions. We have added an explanation why developmental variables were not included into the Methods (paragraph beginning L218).

Third, finding lack of support for a hypothesis in one clade does not negate the hypothesis. This is especially true for brain size studies: some variables are important in some clades and not others. In fact, the originator of the SBH have emphasized this over and over again to explain the lack of concordance between primate data and that of other mammalian clades (as well as birds).

We are careful not to draw any specific conclusions from the results of our models, other than that results are inconsistent depending on methodological choices. Any study claiming strong support for the SIH or EIH based on such methods must therefore be taken with caution. Wartel et al (2019) have also made similar points about comparative analyses of primate brain size, suggesting the issues we highlight are unlikely to be clade-specific. We do not at any point suggest that we support or refute a particular hypothesis.

6. I did not understand how the conclusions of the authors followed logically from their results. For example, what data was presented that indicates that “brain size studies are not methodologically robust”? This statement implies that brain size is not measured properly or that the statistics used are inherently flawed, neither of which are shown by the authors.

As indicated above, the authors do not effectively show the magnitude of intraspecific variation or how that would cause “unreliable results”. In fact, that phrase itself is somewhat inflammatory given the lack of evidence provided by the authors. In addition, the authors emphasize that different modelling approaches caused different results, which is entirely expected and not novel. The fact that the inclusion of different variables and combination of those variables generates different results is a given for any comparative study and does not mean that the subject is unreliable or flawed. It probably means that some variables covary with one another in complex ways, which is to be expected for studies of brain size.

Following this comment, we realise that the word methodologically has ambiguous interpretations, and thank the reviewer for bringing this to our attention. We have changed the wording in the manuscript so that our meaning is clearer (e.g. L27, L328). We do not, however, agree that our conclusions do not follow logically from our results. Indeed, all of the other three reviewers felt our conclusions were justified by our results. We show that there is considerable variation in brain and body size measurements across species, which most likely can be attributed to intraspecific variation. Because most comparative studies of brain size use only one datapoint of brain size and body mass per species, this variation between datasets – no matter the cause – has the capability to influence the results of analyses. This has not been considered in most comparative brain size studies, and highlights one potential reason why the results of such studies may not be robust. Moreover, we show that modelling decisions (e.g. classification of variables and which covariates are included in models) can result in substantially different model results and conclusions. While including different covariates in models will of course lead to different results, the approach used by many researchers (primarily focussing on social or ecological variables while testing the EIH and/or the SIH) leads to strong conclusions for results that may not hold true if other relevant covariates are included. Indeed, this instability may be because of complicated correlations between the variables, which in our opinion is another reason to be sceptical of the results of any such models. The instability of model results has also been highlighted in Wartel et al. (2019), who also advocate that we move away from such methods.

The authors also include a lot of discussion about variables used to test the EIH. This was neither warranted nor necessary.

We do not understand why the reviewer feels that discussion of these variables is not warranted or necessary. We wanted to justify the inclusion of each of our covariates, by showing that and explaining why they are commonly used variables in the field. 

Somewhat bizarrely, the authors finally get to discussing the two hypotheses to state that they should not be considered dichotomous, suggesting that they began the premise of this manuscript with a straw-man argument.

This is not a straw-man argument (and we note that none of the other three reviewers felt it was either). We begin the manuscript by saying many studies treat the SIH and EIH as dichotomous variables, which we provide ample evidence for. After critiquing the methods used in the field, we suggest not only that we move away from these current methods, but that we also move away from treating these hypotheses as if they are dichotomous. We justify this position in the discussion (paragraph beginning L412).

Finally, the end of the Discussion is not what can be taken away from this study. The relationship between social and ecological environment and intraspecific level cognitive performance need not be the same relationship we see across species. Indeed, neuron numbers do not reflect individual performance within mammalian species, but do seem to explain species differences in cognitive performance. 

We do not follow this line of reasoning. Both large-scale comparative studies of brain size evolution and intra-specific studies of cognitive evolution aim to understand the selection pressures that ultimately cause cognition to evolve. This selection occurs at the individual and population-scale. Studying cognitive evolution at this scale will thus give us a more fine-scale and intricate understanding of the driving forces of the evolution of cognition. Comparative studies are of course valuable in understanding broader-scale patterns. However, given the flaws of comparative studies of brain size as a method to understand cognitive evolution, highlighted both here and in e.g. Logan et al., 2018; Powell et al., 2017; Wartel et al., 2019, we advocate for researchers to move toward quantifying the level of uncertainty (and thus information-processing challenges) different species face across different contexts using an intra-specific approach, or by studying closely related taxa. 

Unfortunately, due to the theoretical and analytical issues I have outlined above, I am not sure anything can be taken away from the analyses presented in its current form.

We are sorry that the reviewer feels this way and have endeavoured to clarify our arguments further, but we would point out that this view is at odds with that of the other three reviewers (R1: “Brilliant job! A very intriguing and eye-opening manuscript”; R2: “What a pleasure was to read such an interesting work”; R3: “A well-written and concise text that highlights some fundamental questions about how we study brain size in relation to the evolution of cognition”).

7. Overall, I was also not impressed with the literature cited. Many broad statements were made that cited a cherry-picked subset of papers, while ignoring more recent contributions to the field of research. For example, there are several papers by Jeroen Smaers that emphasize other aspects of brain size evolution, like evolutionary rate changes and allometry, that would have been appropriate to include as part of the Introduction or Discussion, but were not. Similarly, recent studies on intraspecific allometry and the degree of intraspecific variation were also not cited, despite intraspecific variation being the focus of this study. A broader and more careful treatment of the literature is needed.

Thank you for this suggestion. We have included references to two recent allometry studies in the discussion (L374). We have also included citations related to intraspecific variation in brain and body size (L352, L353). 

8. Minor comments

a. Please refrain from using the term “type specimen” in this context. That term specifically refers to the specimen for which a species description/identification is based on.

Thank you. We have changed the wording of “type specimen”.

b. A better figure legend is needed for Figure 1 as it is a bit unclear at first what “species” is referring to on the y-axis.

Thank you. We have changed the wording of Figure 1’s legend (p. 12)

Literature cited

Beauchamp, G., & Fernández-Juricic, E. (2004). Is there a relationship between forebrain size and group size in birds? Evolutionary Ecology Research, 6(6), 833–842.

Burkart, J. M., Hrdy, S. B., & Van Schaik, C. P. (2009). Cooperative breeding and human cognitive evolution. Evolutionary Anthropology, 18(5), 175–186. https://doi.org/10.1002/evan.20222

Chambers, H. R., Heldstab, S. A., & O’Hara, S. J. (2021). Why big brains? A comparison of models for both primate and carnivore brain size evolution. Plos One, 16(12), e0261185. https://doi.org/10.1371/journal.pone.0261185

DeCasien, A. R., Barton, R. A., & Higham, J. P. (2022). Understanding the human brain: insights from comparative biology. Trends in Cognitive Sciences, 1–14. https://doi.org/10.1016/j.tics.2022.02.003

DeCasien, A. R., Williams, S. A., & Higham, J. P. (2017). Primate brain size is predicted by diet but not sociality. Nature Ecology & Evolution, 1(5), 0112. https://doi.org/10.1038/s41559-017-0112

Emery, N. J., Seed, A., Bayern, A. M. P. Von, & Clayton, N. S. (2007). Cognitive adaptations of social bonding in birds. Philosophical Transactions of the Royal Society B, 362(1480), 489–505. https://doi.org/10.1098/rstb.2006.1991

Iwaniuk, A. N., & Arnold, K. E. (2004). Is cooperative breeding associated with bigger brains? A comparative test in the Corvida (Passeriformes). Ethology, 110(3), 203–220. https://doi.org/10.1111/j.1439-0310.2003.00957.x

Logan, C. J., Avin, S., Boogert, N., Buskell, A., Cross, F. R., Currie, A., Jelbert, S., Lukas, D., Mares, R., Navarrete, A. F., Shigeno, S., & Montgomery, S. H. (2018). Beyond brain size: Uncovering the neural correlates of behavioral and cognitive specialization. Comparative Cognition & Behavior Reviews, 13, 55–89. https://doi.org/10.3819/CCBR.2018.130008

Minias, P., & Podlaszczuk, P. (2017). Longevity is associated with relative brain size in birds. Ecology and Evolution, 7(10), 3558–3566. https://doi.org/10.1002/ece3.2961

Overington, S. E., Morand-Ferron, J., Boogert, N. J., & Lefebvre, L. (2009). Technical innovations drive the relationship between innovativeness and residual brain size in birds. Animal Behaviour, 78(4), 1001–1010. https://doi.org/10.1016/j.anbehav.2009.06.033

Powell, L. E., Isler, K., Barton, R. A., Powell, L. E., & Barton, R. A. (2017). Re-evaluating the link between brain size and behavioural ecology in primates. Proc Royal Soc, 1–8.

Pravosudov, V. V., Sanford, K., & Hahn, T. P. (2007). On the evolution of brain size in relation to migratory behaviour in birds. Animal Behaviour, 73(3), 535–539. https://doi.org/10.1016/j.anbehav.2006.10.005

Rosati, A. G. (2017). Foraging Cognition: Reviving the Ecological Intelligence Hypothesis. Trends in Cognitive Sciences, 21(9), 691–702. https://doi.org/10.1016/j.tics.2017.05.011

Sayol, F., Maspons, J., Lapiedra, O., Iwaniuk, A. N., Székely, T., & Sol, D. (2016). Environmental variation and the evolution of large brains in birds. Nature Communications, 7. https://doi.org/10.1038/ncomms13971

Shultz, S., & Dunbar, R. I. M. (2010). Social bonds in birds are associated with brain size and contingent on the correlated evolution of life-history and increased parental investment. Biological Journal of the Linnean Society, 100(1), 111–123. https://doi.org/10.1111/j.1095-8312.2010.01427.x

Sol, D., Garcia, N., Iwaniuk, A., Davis, K., Meade, A., Boyle, W. A., & Székely, T. (2010). Evolutionary divergence in brain size between migratory and resident birds. PLoS ONE, 5(3), 1–8. https://doi.org/10.1371/journal.pone.0009617

Todorov, O. S., Blomberg, S. P., Goswami, A., Sears, K., Drhlík, P., Peters, J., & Weisbecker, V. (2021). Testing hypotheses of marsupial brain size variation using phylogenetic multiple imputations and a Bayesian comparative framework. Proceedings of the Royal Society B: Biological Sciences, 288(1947). https://doi.org/10.1098/rspb.2021.0394

Wartel, A., Lindenfors, P., & Lind, J. (2019). Whatever you want: Inconsistent results are the rule, not the exception, in the study of primate brain evolution. PLoS ONE, 14(7), 1–15. https://doi.org/10.1371/journal.pone.0218655

West, R. (2014). The evolution of large brain size in birds is related to social , not genetic , monogamy social , not genetic , monogamy. Biological Journal of the Linnean Society, 111, 668–678. https://doi.org/10.1111/bij.12193

---

## [Decision Letter · Decision Letter 1]

16 May 2022

PONE-D-22-00405R1Problems with using comparative analyses of avian brain size to test hypotheses of cognitive evolutionPLOS ONE

Dear Dr. Hooper,

Thank you for submitting your manuscript to PLOS ONE. After careful consideration, we feel that it has merit but does not fully meet PLOS ONE’s publication criteria as it currently stands. Therefore, we invite you to submit a revised version of the manuscript that addresses the points raised during the review process.

We look forward to receiving your revised manuscript.

Kind regards,

Vitor Hugo Rodrigues Paiva, Ph.D.

Academic Editor

PLOS ONE

Journal Requirements:

Reviewers' comments:

Reviewer's Responses to Questions

**Comments to the Author**

1. If the authors have adequately addressed your comments raised in a previous round of review and you feel that this manuscript is now acceptable for publication, you may indicate that here to bypass the “Comments to the Author” section, enter your conflict of interest statement in the “Confidential to Editor” section, and submit your "Accept" recommendation.

Reviewer #1: All comments have been addressed

Reviewer #2: All comments have been addressed

Reviewer #3: All comments have been addressed

2. Is the manuscript technically sound, and do the data support the conclusions?

Reviewer #1: Yes

Reviewer #2: Yes

Reviewer #3: Yes

3. Has the statistical analysis been performed appropriately and rigorously? 

Reviewer #1: Yes

Reviewer #2: Yes

Reviewer #3: Yes

4. Have the authors made all data underlying the findings in their manuscript fully available?

Reviewer #1: Yes

Reviewer #2: Yes

Reviewer #3: Yes

5. Is the manuscript presented in an intelligible fashion and written in standard English?

Reviewer #1: Yes

Reviewer #2: Yes

Reviewer #3: Yes

6. Review Comments to the Author

Reviewer #1: Research Article:

1.1 Recommendation

Revise Minor

2. Comments to Author

Ms. Ref. No.: PONE-D-22-00405R1

Title: Problems with using comparative analyses of avian brain size to test hypotheses of cognitive evolution

Overview and general recommendation:

This is coming along nicely! I enjoyed re-reading it and reviewing it again as it has improved so much and reads a lot better. I love the new additions to it and it is much more clear and easier to read. I just have a few comments and suggestions mainly regarding some grammar but other than that I think it is a great paper which can improve the understanding of avian brain size and cognitive evolution.

2.1 Minor comments:

• Please insert a ‘:’ after the italic subheading and before the first word of the sentence in lines 103, 118, 136, 138, 145, 154, 166, 169, 175, 194, 211, 218, 224, 238, 255 and 293.

For example, you could insert the “:” like this in line 136 “Variables: We extracted/collated”.

• Line 105 – Add “s” to bird so it is plural i.e., “species of birds”.

• Line 167 – you end a sentence mentioning “for seven diet types”. Could you please elaborate on those seven diet types and the range of values i.e., is it 1-10 or 1-50.

• Line 170-171 – can you define in briefly “social foraging’ and “cooperative breeding”

• Line 321 – Table 2 – I am assuming that the significant results are in bold. Could you please indicate in the table title if that is correct.

• Line 367 – change “madd” to “mass”

Reviewer #2: Dear authors,

Again, It was a pleasure to read your work. I ‘m glad you have considered all my suggestions and found them helpful in improving your work. I think that you have made a great effort to address all comments and suggestions from the other reviewers and me, and I consider your work will add to the debate on the cognitive evolution in birds. After reading the MS, I found no more issues but recommend you review it again for any minor details.

Congratulations on your great job, first developing such an exciting idea, then improving your MS.

Reviewer #3: I thank the authors for their detailed response and for addressing all comments made. I can see an improvement in the text and especially in the discussion. I also think the feedback from the fourth reviewer helped improve it as well, and I'm glad you were able to catch some of the issues with the dataset in time.

7. PLOS authors have the option to publish the peer review history of their article (what does this mean?). If published, this will include your full peer review and any attached files.

Reviewer #1: No

Reviewer #2: No

Reviewer #3: **Yes: **Martin Chavez-Hoffmeister

---

## [Author Response · Author response to Decision Letter 1]

15 Jun 2022

Reviewer 1

Overview and general recommendation:

This is coming along nicely! I enjoyed re-reading it and reviewing it again as it has improved so much and reads a lot better. I love the new additions to it and it is much more clear and easier to read. I just have a few comments and suggestions mainly regarding some grammar but other than that I think it is a great paper which can improve the understanding of avian brain size and cognitive evolution.

We are so glad you enjoyed re-reading it and feel that it has improved following revision. Thank you for your time in helping us to improve this manuscript. Below we have addressed your specific points in this review round: 

2.1 Minor comments:

• Please insert a ‘:’ after the italic subheading and before the first word of the sentence in lines 103, 118, 136, 138, 145, 154, 166, 169, 175, 194, 211, 218, 224, 238, 255 and 293. 

For example, you could insert the “:” like this in line 136 “Variables: We extracted/collated”.

Good suggestion – we have done this.

• Line 105 – Add “s” to bird so it is plural i.e., “species of birds”.

We rephrased to “bird species”.

• Line 167 – you end a sentence mentioning “for seven diet types”. Could you please elaborate on those seven diet types and the range of values i.e., is it 1-10 or 1-50.

We have now added an explanation of what the seven diet types are, and the range of values in our sample.

• Line 170-171 – can you define in briefly “social foraging’ and “cooperative breeding”

We have now added brief definitions of these terms.

• Line 321 – Table 2 – I am assuming that the significant results are in bold. Could you please indicate in the table title if that is correct.

Yes – we added this to the caption.

• Line 367 – change “madd” to “mass”

Thank you for catching this – it is now corrected.

Reviewer 2

Again, It was a pleasure to read your work. I ‘m glad you have considered all my suggestions and found them helpful in improving your work. I think that you have made a great effort to address all comments and suggestions from the other reviewers and me, and I consider your work will add to the debate on the cognitive evolution in birds. After reading the MS, I found no more issues but recommend you review it again for any minor details.

Congratulations on your great job, first developing such an exciting idea, then improving your MS.

Thank you, we are glad that you enjoyed reading our revised manuscript and thank you for the time you invested in your helpful and insightful review!

Reviewer 3

I thank the authors for their detailed response and for addressing all comments made. I can see an improvement in the text and especially in the discussion. I also think the feedback from the fourth reviewer helped improve it as well, and I'm glad you were able to catch some of the issues with the dataset in time.

Thank you for the time you invested in your detailed and insightful review. We agree that R4’s feedback was also very useful, and are glad that you see an improvement in our manuscript following revisions.

---

## [Editor Report · Decision Letter 2]

17 Jun 2022

Problems with using comparative analyses of avian brain size to test hypotheses of cognitive evolution

PONE-D-22-00405R2

Dear Dr. Hooper,

We’re pleased to inform you that your manuscript has been judged scientifically suitable for publication and will be formally accepted for publication once it meets all outstanding technical requirements.

Kind regards,

Vitor Hugo Rodrigues Paiva, Ph.D.

Academic Editor

PLOS ONE
---

## [Editor Report · Acceptance letter]

28 Jun 2022

PONE-D-22-00405R2 

Problems with using comparative analyses of avian brain size to test hypotheses of cognitive evolution 

Dear Dr. Hooper:

I'm pleased to inform you that your manuscript has been deemed suitable for publication in PLOS ONE. Congratulations! Your manuscript is now with our production department. 

Kind regards, 

on behalf of

Dr. Vitor Hugo Rodrigues Paiva 

Academic Editor

PLOS ONE